# TaskSet: A Dataset of Optimization Tasks

## Abstract

We present TaskSet, a dataset of tasks for use in training and evaluating optimizers. TaskSet is unique in its size and diversity, containing over a thousand tasks ranging from image classification with fully connected or convolutional neural networks, to variational autoencoders, to non-volume preserving flows on a variety of datasets. As an example application of such a dataset we explore meta-learning an ordered list of hyperparameters to try sequentially. By learning this hyperparameter list from data generated using TaskSet we achieve large speedups in sample efficiency over random search. Next we use the diversity of the TaskSet and our method for learning hyperparameter lists to empirically explore the generalization of these lists to new optimization tasks in a variety of settings including ImageNet classification with Resnet50 and LM1B language modeling with transformers. As part of this work we have opensourced code for all tasks, as well as 29 million training curves for these problems and the corresponding hyperparameters.[1]

## 1 Introduction

As machine learning moves to new domains, collecting diverse, rich, and application-relevant datasets is critical for its continued success. Historically, research on learning optimization algorithms have only leveraged single tasks (Andrychowicz et al., 2016; Metz et al., 2019a), or parametric synthetic tasks (Wichrowska et al., 2017), due to the difficulty of obtaining large sets of tasks.

### 1.1 TaskSet: A set of tasks

We present a set of tasks significantly larger than any optimizer dataset previously studied. We aim to better enable standardized research on optimizers, be that analysis of existing optimizers, or development of new learned learning algorithms. We call this suite of tasks TaskSet.

Much in the same way that learned features in computer vision outpaced hand designed features (Krizhevsky et al., 2012; LeCun et al., 2015), we believe that data driven approaches to discover optimization algorithms will replace their hand designed counterparts resulting in increased performance and usability. To this end, standardizing a large suite of optimization tasks is an important first step towards more rigorous learned optimizer research.

In this setting, a single "example" is an entire training procedure for a task defined by data, loss function, and architecture. Thus, TaskSet consists of over a thousand optimization tasks, largely focused on deep learning (neural networks). They include image classification using fully connected and convolutional models, generative models with variational autoencoders (Kingma & Welling, 2013) or flows (Dinh et al., 2016; Papamakarios et al., 2017), natural language processing tasks including both language modeling and classification, as well as synthetic tasks such as quadratics, and optimization test functions. The problems themselves are diverse in size, spanning 7 orders of magnitude in parameter count, but remain reasonably fast to compute as almost all tasks can be trained 10k iterations on a CPU in under one hour. To demonstrate the breadth of this dataset we show an embedding of all the tasks in Appendix A.1 in Figure S1.

---

[1] redacted url

## 1.2 AMORTIZING HYPERPARAMETER SEARCH

Machine learning methods are growing ever more complex, and their computational demands are increasing at a frightening pace (Amodei & Hernandez, 2018). Unfortunately, most modern machine learning models also require extensive hyperparameter tuning. Often, hyperparameter search is many times more costly than the final algorithm, which ultimately has large economic and environmental costs (Strubell et al., 2019).

The most common approach to hyperparameter tuning involves some form of quasi-random search over a pre-specified grid of hyperparameters. Building on past work (Wistuba et al., 2015b; Pfisterer et al., 2018), and serving as a typical example problem illustrative of the sort of research enabled by TaskSet, we explore a hyperparameter search strategy consisting of a simple ordered list of hyperparameters to try. The idea is that the first few elements in this list will cover most of the variation in good hyperparameters found in typical machine learning workloads.

We choose the elements in this list by leveraging the diversity of tasks in TaskSet, by meta-learning a hyperparameter list that performs the best on the set of tasks in TaskSet. We then test this list of hyperparameters on new, larger machine learning tasks.

Although learning the list of hyperparameters is costly (in total we train $\sim$29 million models consisting of over 4,000 distinct hyper parameter configurations), our final published list is now available as a good starting guess for new tasks.

Furthermore, we believe the raw training curves generated by this search will be useful for future hyperparameter analysis and meta-learning research, and we release it as part of this work. We additionally release code in Tensorflow (Abadi et al., 2016), Jax (Bradbury et al., 2018), and Py-Torch (Paszke et al., 2019) for a reference optimizer which uses our learned hyperparameter list, and can be easily applied to any model.

## 2 TASKSET: A SET OF TASKS

How should one choose what problems to include in a set of optimization tasks? In our case, we strive to include optimization tasks that have been influential in deep learning research over the last several decades, and will be representative of many common machine learning problems. Designing this dataset requires striking a balance between including realistic large-scale workloads and ensuring that tasks are fast to train so that using it for meta-learning is tractable. We construct our dataset largely out of neural network based tasks. Our chosen tasks have between ten thousand and one million parameters (much smaller than the billions commonly used today), as a result most problems can train in under an hour on a cloud CPU with 5 cores. We additionally focus on increased "task diversity" by including many different kinds of training algorithms, architectures, and datasets – inspired by past work in reinforcement learning which has demonstrated large numbers of problems and increased diversity around some domain of interest is useful for both training and generalization Heess et al. (2017); Tobin et al. (2017); Cobbe et al. (2018); OpenAI et al. (2019). Again though, a balance must be struck, as in the limit of too much diversity no learning can occur due to the no free lunch theorem (Wolpert & Macready, 1997). Our dataset, TaskSet, is made up of 1162 tasks in total. We define a task as the combination of a loss function, a dataset, and initialization.

Specifically we define a task as a set of 4 functions:

- **Initialization**: () $\rightarrow$ parameter initial values
- **Data generator**: data split (e.g. train / valid / test) $\rightarrow$ batch of data
- **Forward pass**: (batch of data, params) $\rightarrow$ loss
- **Gradient function**: (input data, params) $\rightarrow$ gradients ($\frac{\text{dloss}}{\text{dparams}}$)

A task has no tunable hyperparameters and, coupled with an optimizer, provides all the necessary information to train using first order optimization. This makes experimentation easier, as each task definition specifies hyperparameters such as batch size (Shallue et al., 2018; McCandlish et al., 2018) or initialization (Schoenholz et al., 2016; Yang & Schoenholz, 2017; Xiao et al., 2018; Li & Nguyen,

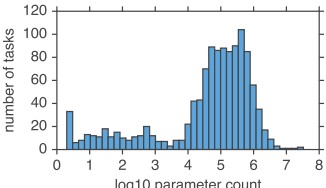 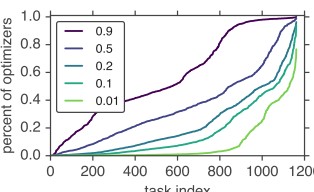 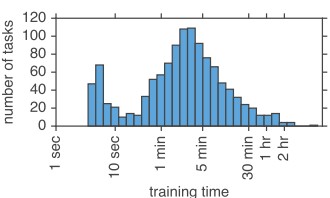

Figure 1: **(a)** A histogram of parameter counts for each problems in the task suite. Our task suite spans more than 7 orders of magnitude in model size. **(b)** Percentage of optimizers (y-axis) capable of reaching a given loss value (color) for tasks (x-axis). We find there exists around 100 "easy" tasks on which more than half of the optimizers perform well, and a large number of "difficult" tasks for which almost no optimizers perform well. **(c)** A histogram of training times. Almost all tasks can be trained in under an hour.

2019; Pretorius et al., 2018; Hayou et al., 2018; Karakida et al., 2018; Blumenfeld et al., 2019; Hayou et al., 2019) that no longer need to be tuned.

We augment a set of "fixed" tasks which have been designed by hand, with "sampled" tasks that are randomly generated task instances.

## 2.1 SAMPLED FAMILIES OF TASKS

Sampled tasks are created by sampling neural network architectures (e.g., MLPs, convnets), activation functions, datasets (e.g., images, text, quadratic functions, and synthetic tasks), and other properties. We organize these sampled tasks into similar *families* of tasks. See Appendix H for a complete description of these sampled tasks. Broadly, these are separated into tasks sampling image models (*mlp, mlp_ae* (Hinton & Salakhutdinov, 2006)*, mlp_vae* (Kingma & Welling, 2013)*, conv_pooling, conv_fc, nvp* (Dinh et al., 2016)*, maf* (Papamakarios et al., 2017)), tasks sampling language models (*char_rnn_language_model* (Graves, 2013)*, word_rnn_language_model, rnn_text_classification*), quadratics (*quadratic*) and other synthetic tasks (*losg_tasks* (Wichrowska et al., 2017)). Defining a sampling distribution that generates tasks that are always valid, and that run within a time constraint, is difficult. Instead, we define a broad distribution and make use of rejection sampling to remove tasks that are either too slow or that we are unable to optimize at all. By starting with a distribution that is too broad, and pruning it, we hope to achieve better coverage of tasks.

## 2.2 HAND DESIGNED TASKS

In addition to the sampled tasks, we also include 107 hand designed tasks. These consist of more common tasks that both improve the coverage beyond the sampled tasks, and provide for better interpretability through a closer match to existing tasks in the literature. These tasks span image classification, text classification, language modeling, and generative modeling, as well as some synthetic tasks such as associative retrieval (Ba et al., 2016). We leave the description of each one of these tasks to Appendix H.3.

## 2.3 AGGREGATE STATISTICS OF TASKSET

In Figure 1a we show histograms of compute times for all problems and find almost all problems train under an hour (see Appendix C for per task family histograms). In Figure 1c we plot a histogram of the number of parameters per tasks. Finally, in Figure 1b we show a distribution of task difficulty by plotting the fraction of optimizer configurations that achieve a certain loss value. We find that for some tasks as many as $50\%$ of optimizers perform well while for others $< 1\%$ achieve a loss close to the smallest observed loss. For a qualitative visualization of TaskSet, see Appendix A

## 3 AMORTIZED HYPERPARAMETER SEARCH

As a simple demonstration of using TaskSet for meta-learning research, we consider learning hyper-parameter lists. This idea of learning lists of hyper parameters has been explored in (Wistuba et al.,

2015b; Pfisterer et al., 2018). We define an optimizer as the pairing of an optimization algorithm and all its corresponding hyperparameters (e.g. learning rate). While sometimes practitioners use a single optimizer – e.g. Adam (Kingma & Ba, 2014) with default hyperparameters – most practitioners will often run multiple optimizers and use a validation set to select the best performer.

## 3.1 OPTIMIZER FAMILIES

We define different parameterizations of hand designed optimizers as an optimizer family. The optimizer families we consider consist of:

- *Adam1p*: One hyperparameter, the fixed learning rate $\alpha$
- *Adam4p*: Four Adam hyperparameters, $\alpha$, $\beta_1$, $\beta_2$, and $\epsilon$
- *Adam6p*: Adam4p hyperparameters, and two additional hyperparameters controlling linear and exponential learning rate decays
- *Adam8p*: The hyperparameters in *Adam6p* plus two additional hyperparameters for $\ell 1$ and $\ell 2$ regularization terms
- *NAdamW*: A 10 hyperparameter search space based on NAdam (Dozat, 2016) with cosine learning rate decay, and weight decay.

For the full update equations see Appendix D.1 for Adam and D.2 for NadamW. We chose Adam based on its use in existing work, and NAdam based on performance shown in (Choi et al., 2019).

## 3.2 LEARNED HYPERPARAMETER LISTS

Traditionally researchers tune hyperparameters on a per model basis. While this often results in performance gains; it comes at the cost of immense compute, and researchers are almost never able to expend enough compute to saturate model performance (Shallue et al., 2018). As an alternative to per-problem tuning, we proposes instead tuning the search strategy itself on a dataset of tasks and *transferring* the knowledge gained to new tasks of interest. This idea is already implicitly done by humans – e.g. we don't start a hyperparameter search with a learning rate of $10^6$ – we use values that the community has found useful.

This dataset-based tuning has a number of desirable properties. First, the resulting search strategies are much more efficient, resulting in large speedups in sample efficiency on unseen tasks over a random search baseline. Second, we are less restricted by the number of optimizer parameters we search over or by needing to define reasonable search spaces. For example, if there are redundant regions of search space, our learned optimizer will be less likely to sample them repeatedly, unlike random search. If there is a region of hyperparameter space that performs poorly on all problems, the learned search strategy will avoid it.

In this work we parameterize the learned search strategy as an ordered list of optimizers to try (i.e. a list of hyperparameter configurations). Given a fixed number of task evaluations we would like to achieve the best possible performance on all tasks in the training set of tasks. For a length $k$ list of optimizers we define our loss as:

$$J(\theta_{1,...,k}) = \sum_{\tau \in \text{tasks}} \left[ \min_{i \in 1..k} f(\tau, \theta_i) \right], \tag{1}$$

where $\theta_i$ are the optimizer hyperparameters for element $i$ in the list, and $f$ is an appropriately normalized loss computed after training task $\tau$.

We seek to find an optimal list of optimizers as (similar to (Wistuba et al., 2015b)):

$$\theta_{1,...,k}^* = \underset{\theta_{1,...,k}}{\arg\min} J(\theta_{1,...,k}). \tag{2}$$

This is meant to serve as an example task, illustrative of the sort of research enabled by TaskSet. More advanced hyperparameter search strategies would no doubt yield even more performant results.

### 3.3 SCORING AN OPTIMIZER BY AVERAGING OVER TASKS

To score a task, we initialize the parameters of the task and run 10,000 iterations of an optimizer. We monitor loss on each data split (train, validation, test) every 200 steps using an average over 50 mini-batches per evaluation. For all data presented in this paper we also compute averages over 5 random task parameter initializations.

A side effect of the diverse task dataset is that losses span multiple orders of magnitude, making direct aggregation of performance problematic. To remedy this we normalize the loss values for all tasks linearly between 0 and 1 where 1 is validation loss at initialization and zero is the lowest validation loss achieved by any tested optimizer. Loss values greater than the loss at initialization are clipped to 1. To collapse an entire normalized training curve into a scalar cost, we compute the mean normalized loss over the 10,000 iterations. We find empirically that this choice is similar to taking the minimum (Appendix B.5). We leave exploring alternative methods such as performance profiles (Dolan & Moré, 2002) and Nash averaging (Balduzzi et al., 2018) for future work.

### 3.4 GREEDY LEARNING FROM RANDOM SEARCH

Optimizing Eq. 2 is combinatorially expensive. To tractably solve this optimization problem, we introduce two approximations (Wistuba et al., 2015b). First, we shift the unconstrained search over the full space of optimizers to search over a finite set of optimizers, $\Theta$. This finite set can be computed ahead of time and decouples the expensive procedure of training each task with an optimizer from training the learned search space. Separating data and training in this way has been done for both hyperparameter search (Eggensperger et al., 2015), and neural architecture search (Klein & Hutter, 2019; Ying et al., 2019). In total we trained 1,000 optimizer configurations for each of Adam1p, Adam4p, Adam6p, Adam8p, and NAdamW on all 1,162 tasks with 5 random seeds per pair. Second, we use a greedy heuristic to approximate the combinatorial search over sets of $k$ optimizers. For a single optimizer trial, $k = 1$, we select the best performing optimizer on average across all training tasks. We then continue to select optimizer parameters such that the minimum of all optimizer-parameters per task, aggregated over all tasks is minimized. This shifts the complexity from exponential in $k$ to linear. Finding a length $k$ set of optimizers can thus be efficiently computed as follows:

$$\theta_1^* = \underset{\theta \in \Theta}{\arg\min} \left[ \sum_{\tau \in \text{tasks}} f(\tau, \theta) \right] \tag{3}$$

$$\theta_k^* = \underset{\theta \in \Theta}{\arg\min} \left[ \sum_{\tau \in \text{tasks}} [\min(b, f(\tau, \theta))] \right] \qquad \text{where} \quad b = \underset{i \in 1..(k-1)}{\min} f(\tau, \theta_i^*). \tag{4}$$

We note that the first argument of the outer min, $b$, can be computed once per set of hyperparameters as it does not depend on $\theta$. Finally, as our tasks are stochastic, we order optimizers based on validation loss and report test loss (Van Hasselt et al., 2016).[2]

This training strategy requires an original search space from which to collect data and build $\Theta$. The search space we use is described in Appendix E.2. While large, we find that the optimal parameters for each task end up covering almost the entire space. At some point, no improvement can be obtained on any of the tasks in the dataset. At this point, we simply randomly order the remaining optimizers though expect more sophisticated methods could be employed.

## 4 EXPERIMENTS: TRAINING AND GENERALIZATION OF LEARNED HYPERPARAMETER LISTS

With our dataset of tasks and data collected, we turn our attention to exploring training of the hyperparameter lists, and generalization beyond the suite of tasks in TaskSet. In this exploration,

---

[2]This technically means that increasing the number of optimizes could potentially decrease performance, but we find this rarely happens in practice.

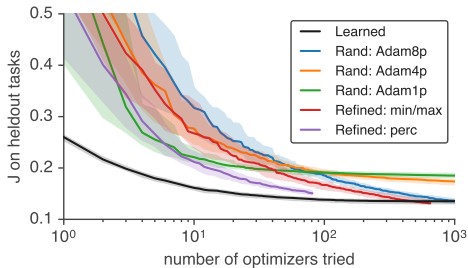

Figure 2: By learning a search space we achieve large speedups over random search. On the y-axis we show J, or the best aggregated and normalized performance achieved given some number of optimizer trials (x-axis). This is computed on heldout tasks not used to train the hyperparameter list. In solid we show median performance with 25-75 percentile shown with error bars over 50 resamplings of the train-test split of tasks, as well as random samplings. In black we show a learned search space computed from the Adam8p family of optimizes. In color we show various random search baselines. See §4.1 for description of these.

we hope to give a flavor of the types of research possible with TaskSet. Our main tool to show performance are figures that sweep the number of optimizers configurations on the x-axis, and show the best performance achieved for each number of optimizers tried, averaged over some set of tasks (Eq. 1).

## 4.1 LEARNED HYPERPARAMETER LISTS ARE MORE EFFICIENT THAN RANDOM SEARCH

To demonstrate the impact of learning a search space, we take the 1,162 tasks split them into even train and test tasks. We then learn a search strategy using optimizers from the Adam8p family following Eq. 4 on the train tasks. Results in Figure 3. As baselines, we use random search with different search spaces, including just learning rate (Rand: Adam1p), the default Adam hyper parameters (Rand: Adam4p), as well as the Adam 8 dimensional search space (Rand: Adam8p). To better get a sense of performance, we show two additional "Refined" baselines which involve random sampling from better search space. For min/max, we sample from the minimum bounding box containing the best hyperparameters for each task. To improve the search space quality, we shrink this bounding box so 90% of the best hyperparameters are enclosed. Further considerations regarding search space volume are treated in E.1, and the precise search spaces are specified in Appendix E.2. Finally, one difficulty of working with offline data is the difficulty of running online hyperparameter optimization methods such as Bayesian Optimization without running additional compute. Future work will explore offline Bayesian methods.

## 4.2 MORE TASKS LEAD TO BETTER GENERALIZATION

We next look at the effects of the number of training tasks on generalization. We take subsets of tasks of different size, and train hyperparameter lists using Eq.4. We compute test performance on the remainder of the tasks and plot loss averaged over different splits in Fig. 3. We find that a large number of tasks (more than 100) are required to achieve near-optimal test performance. This is surprising to us given how simple our learned search strategy is (simply a list of hyperparameters), but not wholly so given past work studying generalization in RL (Cobbe et al., 2018).

## 4.3 GENERALIZATION TO DIFFERENT TYPES OF PROBLEM

For learned algorithms to be generally useful, some amount of generalization to unseen task *families* is required. To test this, we split our data into disjoint task types. We perform two splits: testing on RNN tasks and training on all others, and testing on autoencoder tasks and training on all others. As a best case baseline we additionally train search spaces on the test task families directly. We find an order of magnitude better sample efficiency than random search for both cases and find our learned search space is close in performance to search spaces trained on just the testing tasks (Fig. 3).

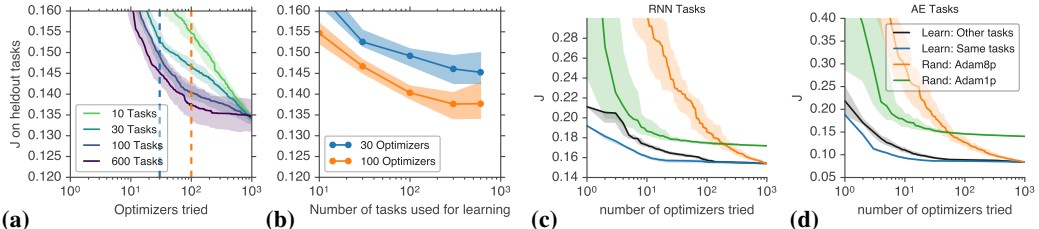

Figure 3: Using more tasks to train the search space results in improved performance on heldout tasks. **(a)** The number of optimizers tried vs performance on heldout tasks. In color, we show different numbers of tasks used to learn the search spaces. We show median performance with error bars denoting 25 and 75 percentile. **(b)** Performance at a fixed number of optimizers tried vs number of tasks used for meta-training. We find performance continues to improve as we meta-train on more tasks. These plots are slices out of (a), with colors matching the vertical dashed lines. **(c,d)** We show aggregate performance (J) as a function of number of optimizers tried when training the hyperparameter list on a different distributions of tasks than those we test on. We show testing on RNNs **(c)** and auto encoders **(d)**, and train on 700 tasks sampled from the remainder set of tasks. We find that these learned search spaces perform much better than random search in both the learning rate search space and in the original Adam8p search space. We additionally plot the best case performance – the case where we train and test on the same problem type. We show median and 25-75 percentile averaged over 50 different samplings.

## 5 EXPERIMENTS: REALISTIC PROBLEMS

In §4.3 and §B.1 we explored generalization of learned hyperparameter lists to held out tasks within the TaskSet dataset. While useful for analysis, these tasks are still far from the workloads commonly employed to solve real problems. In this section, we explore the performance of our learned search space on a number of state of the art models. These models drastically differ from the training set of tasks in parameter count and compute cost. We see these experiments as evidence that the tasks presented in TaskSet capture enough of the structure of "realistic" problems that TaskSet can be used to improve larger scale workloads. For all experiments in this section we take the optimizer ordering using the NAdamW optimizer family on all TaskSet tasks then apply the resulting search space to the target problem. The final ordered list of hyperparameters used is in Appendix G. We show results for ResNet50 on ImageNet, and Transformers on LM1B. Additional results with reinforcement learning using PPO are in Appendix B.2.

First we explore ImageNet classification using a ResNet50. on We take the TPU implementation with default settings from the official Tensorflow models repository (Tensorflow, 2019), and swap out different optimizers. We show accuracy computed over the course of training as well as best performance for a given hyperparameter budget in Figure 4. We find that the learned search space vastly outperforms learning rate tuned Adam.

Next we explore language modeling on LM1B with a Transformer. We take the transformer (Vaswani et al., 2017) example implemented in Jax (Bradbury et al., 2018) with Flax (Flax Developers, 2020). We train using a 2x2 TPU V2 configuration for 100k iterations. Once again we take all other hyperparameters as is and simply swap optimizer implementation. We find the learned hyperparameter list dramatically outperforms the default optimizer setting and the fixed learning rate baseline. Nevertheless, we emphasize that our method does not require any knowledge of the underlying problem to achieve faster results. See Appendix B.3 for this same transformer with a budget of 20k iterations.

## 6 RELATED WORK

The idea of sets of tasks has been explored throughout machine learning. The majority of these suites are for use in evaluation where as our suite is targeted for meta-learning. The closest family of optimization tasks for evaluation to those presented here is DeepObs (Schneider et al., 2019) which

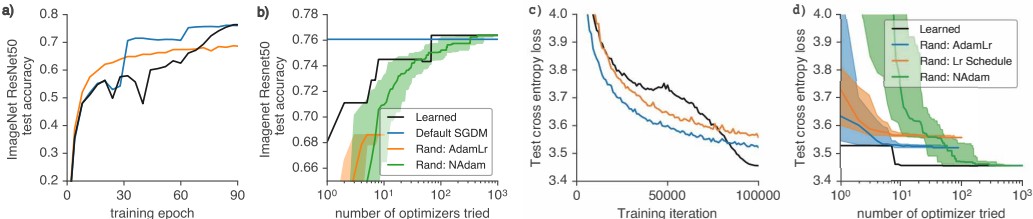

Figure 4: We find our learned optimizer list outperforms both learning rate tuned Adam and default training hyperparameters for ResNet50 and a 53M parameter Transformer trained on LM1B. **a)** Learning curves for the best hyperparameter from each optimizer class on the ResNet50 task. **b)** Number of optimizers tried vs best top 1 ImageNet accuracy achieved on the ResNet50 task. For NAdamW, we first train models with a validation set created by splitting the training set. We compute maxes over this set, and show the performance when retraining on the full training set on the official ImageNet validation set. For AdamLR, we simply compute a max over the official ImageNet validation set. **c)** Training curves for the best hyperparameter for each optimizer class on the Transformer task. **d)** Number of optimizers tried vs the test loss obtained given the best validation performance on the Transformer task.

includes 20 neural network tasks. Our task suite focuses on smaller problems and contains 50x more tasks. Outside of evaluation, task suites in reinforcement learning such as Obstacle Tower (Juliani et al., 2019), ProcGen (Cobbe et al., 2019), CoinRun (Cobbe et al., 2018), and Sonic (Nichol et al., 2018) focus on training algorithms that work across a variety of settings.

The creation of TaskSet was motivated by the goal of learning learning algorithms, or meta-learning (Schmidhuber, 1987; 1995; Hochreiter et al., 2001), and in particular learned optimizers (Bengio et al., 1990; Andrychowicz et al., 2016; Bello et al., 2017; Wichrowska et al., 2017; Li & Malik, 2017; Lv et al., 2017; Metz et al., 2019a;b). This use case is explored with this dataset in (Metz et al., 2020). In this work we do not use this task suite to train learned optimizers, but instead focus on learning a hyperparameter search strategy. Tuning hyperparameters by leveraging multiple tasks has been explored within the contexts of Bayesian optimization Swersky et al. (2013); Perrone & Shen (2019); Perrone et al. (2018) as well as meta-learning (Reif et al., 2012; Gomes et al., 2012; Feurer et al., 2014; Wistuba et al., 2015b;a; Chen et al., 2017; Pfisterer et al., 2018). See Appendix F.1 for a full discussion of sets of tasks in machine learning, Appendix F.2 for more info on optimization in machine learning, and Appendix F.3 for a discussion on existing hyper parameter search methods.

## 7 DISCUSSION

Learning optimization algorithms represents a promising direction for accelerating machine learning research. For the resulting algorithms to become useful tools, however, we must further understand the relationships between training tasks, meta-optimization, and both iid and out of distribution generalization.

This work takes steps towards this goal by introducing a significantly larger set of optimization tasks than ever previously considered. As an example use-case, we provide a thorough analysis of how TaskSet enables meta-optimization of simple, but performant hyperparameter lists. Despite this approach's simplicity, the training of learned learning algorithms is computationally expensive. We hope to explore alternative parameterizations which will increase efficiency by, e.g., leveraging previous evaluations or partial model training (Swersky et al., 2014; Li et al., 2016).

We are releasing the optimal hyperparameter list we have found as a drop-in replacement optimizer in a variety of deep learning frameworks (Tensorflow (Abadi et al., 2016), PyTorch (Paszke et al., 2019), and JAX (Bradbury et al., 2018)) in the hopes that the research community finds them useful. We believe this represents a new set of reasonable optimizer defaults for new problems. Finally, we hope TaskSet encourages more standardized research on general purpose optimizers.

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

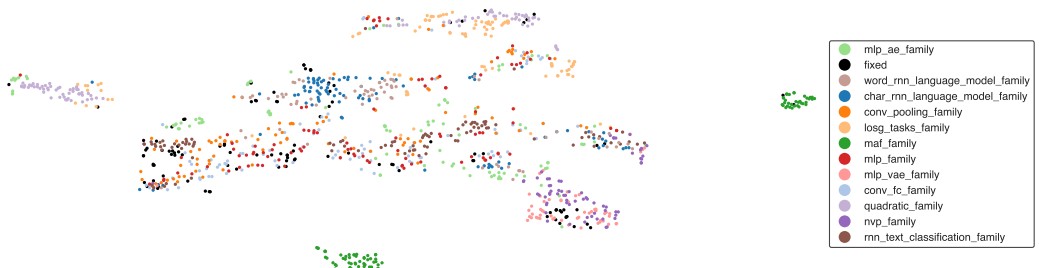

Figure S1: A 2D TSNE embedding of all 1162 tasks. This embedding is produced from a 1,000 dimensional feature vector consisting of task loss evaluated with many different hyperparameter configurations. We find similar tasks – e.g. masked auto regressive flow models, and character / word RNN models – cluster, suggesting similarity in the optimizers that perform well. See §**??** for more details.

## A    TASKSET VISUALIZATION

For a qualitative view, we constructed a feature space consisting of performance measurements for each task+optimizer pair (See §3.3). This forms a dense matrix of size number of tasks by number of optimizers. We then perform T-SNE (Maaten & Hinton, 2008; Van Der Maaten, 2014) to reduce the dimensionality to two and plot the results coloring by task family (Figure S1). Clusters in this space correspond to tasks that work well with similar optimizers. We find diversity of tasks with clusters occurring around similar families of tasks.

### A.1    TSNE OF TASKSET

## B    ADDITIONAL EXPERIMENTS

### B.1    GENERALIZATION TO DIFFERENT SIZED PROBLEMS

Training learned algorithms on large models is often infeasible for computational reasons. As such, one form of generalization needed when building learned algorithms is the ability to transfer to different sized models. As shown in Figure 1 the tasks in this suite contain a wide range of parameter counts, and can thus be used to test this kind of generalization. We split the tasks into 8 groups – one group per order of magnitude in parameter count, and train hyperparameter lists on one range and test on the rest. In Figure S2 we plot the fraction of the training loss achieved by the test loss on the target parameter range. We find peak performance around the model sizes used for training, and smooth falloff as the testing tasks become more dissimilar as measured by parameter count. We note that our problems are not evenly distributed across these groups thus each group will contain a different percentage of the underlying tasks. While this potentially confounds these results, we believe a similar bias occurs in realistic workloads as well.

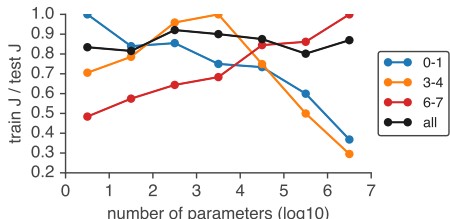

Figure S2: We show learned search space generalization, measured as a ratio of the loss achieved in training and testing, versus the number of task parameters used during search space training. Generalization falls off as one moves further away from the training regime. In black we show that a uniform mixture of the 7 parameter buckets does not fall off.

## B.2 REINFORCEMENT LEARNING WITH PPO

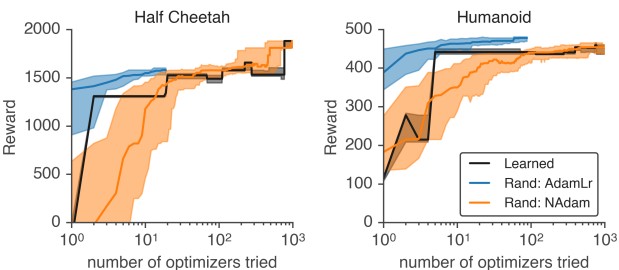

Figure S3: We find our learned hyperparameter lists performs about as well as random search on the NAdam search space, and worse than the random search on the learning rate tuned Adam search space.

We test the learned hyperparameter lists on two continuous control reinforcement learning environments, half cheetah and humanoid, from Gym's Mujoco environments(Todorov et al., 2012; Brockman et al., 2016). We use TF-Agents (Guadarrama et al., 2018) with all non-optimizer hyperparameters set via searching a mixture of environments. In figure B.2 we find our learned hyperparameter lists achieves comparable to slightly worse performance does not out perform learning rate tuning of Adam in both efficiency nor final performance. To diagnose this behavior we ran all 1k optimizers for both problems and found the learned hyperparameter list performs comparable to random search in the underlying space. To probe further, we computed spearman correlation on the performance of each optimizer as compared to the rest of the tasks in the task suite. We found considerably worse correlations than where present for tasks in the TaskSet. This is not surprising as TaskSet contains no reinforcement learning problems.

## B.3 LM1B TARGETING 20K ITERATIONS

We show a transformer on LM1B similar to that shown in §5 except run for only 20k iterations, a fith of the steps. Results in Figure S4. We find the learned hyperparameter lists are much more efficient than either of the baselines.

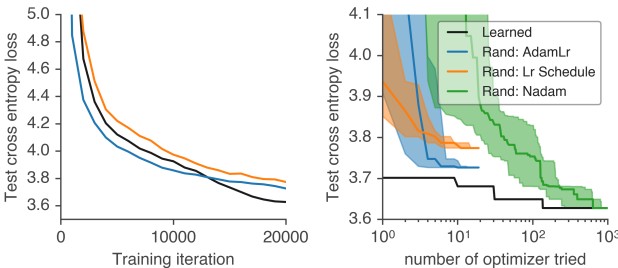

Figure S4: We find our learned hyperparameter lists out performs learning rate tuned Adam with both a constant, and a fixed learning rate schedule on a 53M parameter Transformer trained on LM1B. **Left:** Learning curves for the best of the optimizers. **Right:** Number of optimizers tried vs best test loss.

## B.4 PROBING SHORT HORIZON

Often the goal when training a learned optimizers is to minimize performance after training some number of iterations. This is extremely computationally expensive and in practice approximations must be used. One common family of approximations is short horizon based methods. These methods rely upon somehow truncating training so that updates can be made to the learned optimizer more frequently. This is commonly done via truncated backprop (Werbos, 1990; Wichrowska et al., 2017;

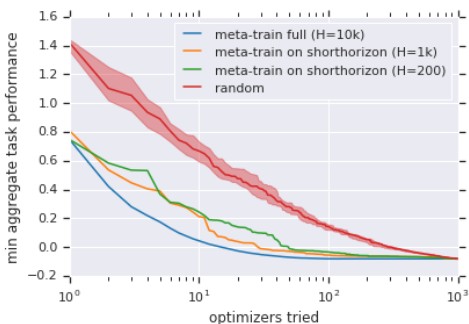

Figure S5: Hyperparameter lists trained on short horizon data generalize remarkably well. On the y-axis we show performance evaluated on the the full 10k training iterations for a given number of optimizers tried (x-axis). In color we show different number of steps used when evaluating task optimizer performance when training the hyperparameter list.

Metz et al., 2019a; Wu et al., 2016), or proxy objectives such as only training for a handful of epoch (Zoph & Le, 2017). While this short horizon proxy is certainly not optimal(Wu et al., 2016), the performance gains are immense and in practice is what makes meta-training optimizers feasible. In our task suite, we test this short horizon learning by training hyperparameter lists only using some finite amount of training iterations per task and testing in the full training regieme (10k steps). Results in figure S5. We find that even when learning the hyperparameter list on a mere 200 steps, our hyperparameter list continues to generalize to outperform random search on Adam8p. This is promising as this suggests that training the learned hyperparameter list can be done with 1/50th of the total compute. This result is surprising to us as prior work indicates the effect of this bias can be severe (Wu et al., 2016; Metz et al., 2019a). We suspect it is due to the simplicity of the learned parameter space but leave a thorough analysis of this for future work.

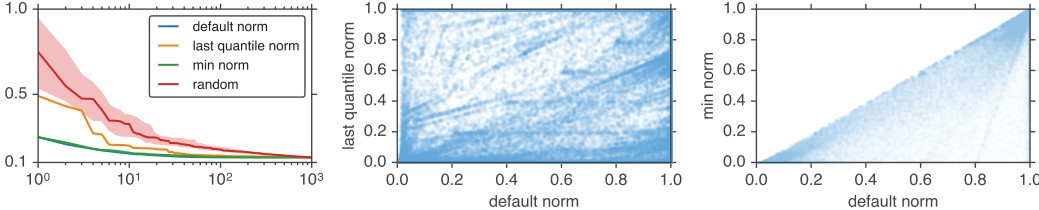

Figure S6: **Left:** Aggregate performance (y-axis) vs number of optimizer tried (x-axis) for different normalization and aggregation techniques. In each curve we train the hyperparameter list with a different normalization and aggregation strategy and test with the default normalization and aggregation technique described in 3.3. We find some some strategies are near identical in performance (e.g. min norm), while others perform significantly worse – e.g. last quantile norm. In both cases, however, we still perform better than the underlying random search. **Center:** Correlation between default normalization and the quantile based normalization strategy. Correlation is quite low – 0.193 Pearson's correlation. **Right:** Correlation between the default normalization using a mean to aggregate over validation over the course of training vs using a min over validation over the course training. We find a much higher correlation of 0.911.

## B.5 CHOICE OF NORMALIZATION FUNCTION

There is no easy way to define a single metric for optimizer performance over a mixture of tasks. This paper picks a single normalization strategy based on minimum validation loss and the validation loss at initialization presented in §3.3. In this section we show the impact of choosing a different normalization and or aggregation technique. First, instead of computing the mean over learning curves as described in §3.3 we compute a min. Second, instead of rescaling based on init and min,

we linearly rescale based on the 95 percentile of validation loss and the min validation loss achieved at the end of training each task.In Figure S6 we show learned hyperparameter list training and testing performance as a function of number of optimizers tried when training with different normalization techniques. We find using the min instead of mean results in a negligible change, while using the percentile loss more significantly hurts performance. This difference can be explained by Figure S6b and S6c where we show correlations between the two losses. We find the percentile loss has a much weaker correlation to the default normalizer. We suspect this difference is due to the fact that many optimizers diverage on tasks. By using the 95 percentile we upweight optimizers that do not diverge.

### B.6 TASK FAMILIES ARE DIVERSE

To show the effects of diversity we train and test hyperparameter lists on each pair of task family. We additionally normalize each column from 0-1 to account for different mean losses across tasks. Results in Figure S7. While we do find some similarity in tasks – e.g. between MAF and NVP models, but no two tasks behave the same performance characteristics (no duplicate columns) suggesting that each task family is providing a different contribution to the space of all tasks. We also find when training on certain "far away" tasks, e.g. the quadratic family, we find poor performance on most other task families.

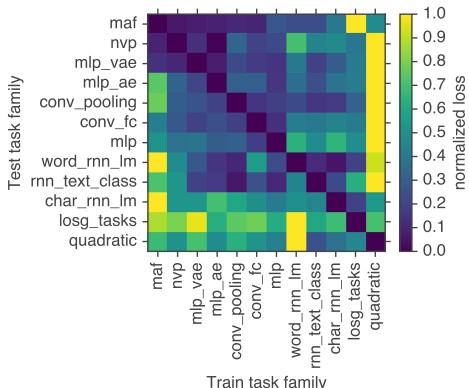

Figure S7: Learning hyperparameter lists using one task family and testing on the remainder of task families. We normalize each column from 0-1 to account for different mean losses across tasks. Lower loss means better performance. We find some groups of similar tasks, but in general no two task families behave identically.

### B.7 EFFECTS OF THE META-TRAINING SEARCH SPACE SIZE

Our offline learning technique described in §3.4 hinges on a finite set of optimizers collected via random search. This set is denote by $\Theta$ in Eq.4. In this section we probe the impact of this size. We take different sized subsets of the the thousand Adam8p optimizer configurations and train and test search spaces on different iid splits of tasks. We then plot performance as a function of this number of optimizers in Figure S9. Moving left in this figure corresponds to increasing the compute needed to train the learned hyperparameter list. We find performance continues to improve as the size of $\Theta$ grows. Given the high dimension of our meta-parameters, 8, this is not a surprise as the number of evaluations needed to explore the space will grow exponentially. We find that the full thousand trials are needed to out perform learning rate tuned Adam when only given a single optimizer evaluation. We find around 100 optimizers (size of $\Theta$) are needed in the case of 10 optimizer trials ($k = 10$).

Overall this suggests that randomsearch might not be the most efficient learning method for creating hyperparameter lists. This is especially true as we work with optimizer families that have more hyperparameters. Other approximate learning methods should likely be explored such as truncated backprop through time as used by the learned optimizer community(Metz et al., 2019a), and/or population based methods (Balduzzi et al., 2019).

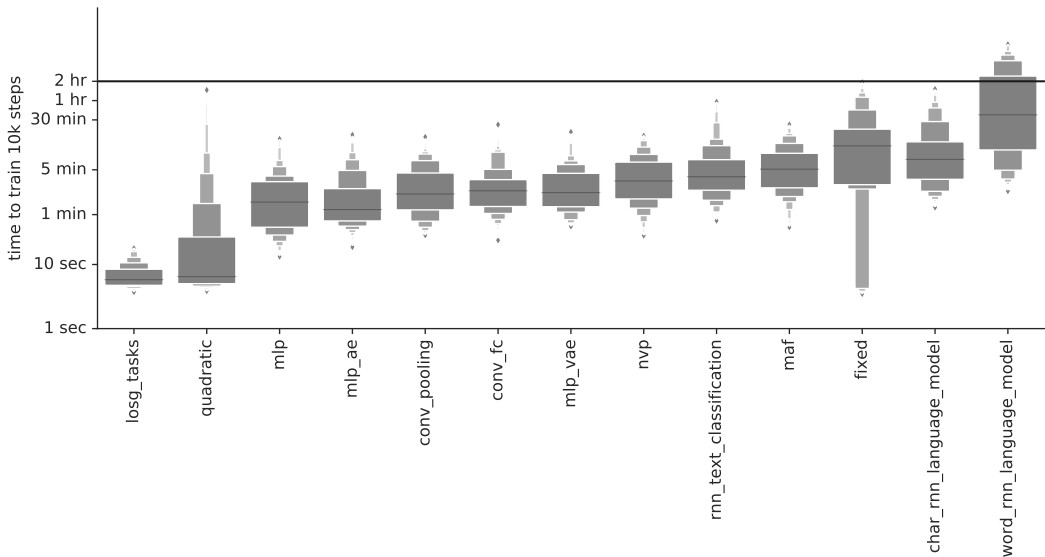

Figure S8: Timings computed for each task family. We find most task families have a narrow distribution of compute times.

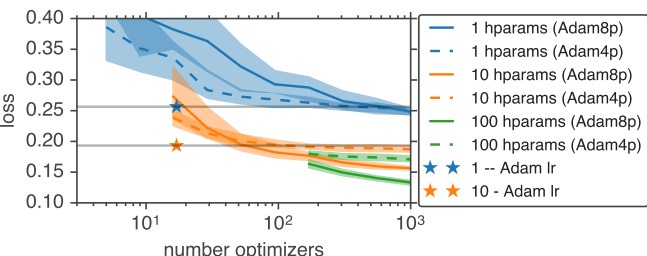

Figure S9: Performance continues to improve as more and more optimizers are used when training the search spaces. On the x-axis we show number of optimzers (size of $\Theta$, the number of hyperparameter evaluations used in training the learned hyperparameter list) and y-axis we show test loss achieved when applying the learned search space for a given fixed length, e.g. different values of $k$ shown in color). We plot median with 25-75 percentile shaded over different random optimizer samples and iid task splits. Stars (with horizontal guide lines) denote best search for the corresponding number of hyperparameters for learning rate tuned Adam in half orders of magnitude.

## C  TASK TIMINGS

In Figure S8 we show box plots of training times for each problem. For each task we use the median step time recorded over a mixture of different physical devices and multipled by 10k to estimate a full training time. Future versions of this dataset of tasks will contain more variation within each task family.

# D    Optimizer family update equations

## D.1    Adam8p update equations

The 8 meta-parameters are: the learning rate, $\alpha$, first and second moment momentum, $\beta_1$, $\beta_2$, the numerical stability term, $\epsilon$, $\ell_2$ and $\ell_1$ regularization strength, and learning rate schedule constants $\lambda_{\text{exp\_decay}}$ and $\lambda_{\text{linear\_decay}}$. For Adam6p, we set $\ell_1$ and $\ell_2$ to zero.

$$\phi^{(0)} = \text{problem specified random initialization} \tag{S1}$$

$$m^{(0)} = 0 \tag{S2}$$

$$v^{(0)} = 0 \tag{S3}$$

$$g^{(t)} = \frac{d}{d\phi^{(t)}}(f(x;\phi^{(t)}) + \ell_2||\phi^{(t)}||_2^2 + \ell_1||\phi^{(t)}||_1) \tag{S4}$$

$$m^{(t)} = \beta_1 m^{(t-1)} + g^{(t)}(1 - \beta_1) \tag{S5}$$

$$v^{(t)} = \beta_2 v^{(t-1)} + (g^{(t)})^2(1 - \beta_2) \tag{S6}$$

$$\hat{m}^{(t)} = \frac{m^{(t)}}{1 - \beta_1^{t+1}} \tag{S7}$$

$$\hat{v}^{(t)} = \frac{v^{(t)}}{1 - \beta_2^{t+1}} \tag{S8}$$

$$u^{(t)} = \frac{\hat{m}^{(t)}}{\sqrt{\hat{v}^{(t)}} + \epsilon} \tag{S9}$$

$$s_{\text{linear}}^{(t)} = \max(1 - t\lambda_{\text{linear\_decay}}, 0) \tag{S10}$$

$$s_{\text{exp}}^{(t)} = \exp(-t\lambda_{\text{exp\_decay}}) \tag{S11}$$

$$\phi^{(t+1)} = \alpha s_{\text{linear}}^{(t)} s_{\text{exp}}^{(t)} u^{(t)} \tag{S12}$$

## D.2    NAdamW update equations

This optimizer family has 10 hyper parameters. The base learning rate, $\alpha_{base}$, first and second moment momentum, $\beta_1$, $\beta_2$, the numerical stability term, $\epsilon$, $\ell_{2WD}$ $\ell_2$ regularization strength, $\ell_{2AdamW}$ AdamW style weight decay, and a boolean to switch between NAdam and Adam, $b_{\text{use nesterov}}$. The learning rate schedule is based off of a single cycle cosine decay with a warmup. It is controlled by 3 additional parameters – $c_{\text{warmup}}$, $c_{\text{constant}}$, and $c_{\text{min learning rate mult}}$.

The learning rate is defined by:

$$u = c_{\text{warmup}}T > t \tag{S13}$$

$$\alpha_{\text{decay\&constant}} = (\alpha_{base} - c_{\text{min learning rate mult}})(0.5 \tag{S14}$$

$$\cos(t\pi/(T - c_{\text{constant}})) + 0.5) + \tag{S15}$$

$$c_{\text{min learning rate mult}} \tag{S16}$$

$$\alpha_{\text{warmup}} = \frac{t}{(T c_{\text{warmup)}}} \tag{S17}$$

$$\alpha = (1 - u)\alpha_{\text{decay\&constant}} + u\alpha_{\text{warm}} \tag{S18}$$

The update equations of NAdamW are quite similar to that of Adam8p. For clarity we list the full update here.

$$\phi^{(0)} = \text{problem specified random initialization} \tag{S19}$$

$$m^{(0)} = 0 \tag{S20}$$

$$v^{(0)} = 0 \tag{S21}$$

$$g^{(t)} = \frac{d}{d\phi^{(t)}}(f(x; \phi^{(t)}) + \ell_{2wd}||\phi^{(t)}||_2^2 \tag{S22}$$

$$m^{(t)} = \beta_1 m^{(t-1)} + g^{(t)}(1 - \beta_1) \tag{S23}$$

$$v^{(t)} = \beta_2 v^{(t-1)} + (g^{(t)})^2(1 - \beta_2) \tag{S24}$$

$$\hat{m}^{(t)} = \frac{m^{(t)}}{1 - \beta_1^{t+1}} \tag{S25}$$

$$\hat{v}^{(t)} = \frac{v^{(t)}}{1 - \beta_2^{t+1}} \tag{S26}$$

$$u_{\text{heavy ball}}^{(t)} = \frac{\hat{m}^{(t)}}{\sqrt{\hat{v}^{(t)}} + \epsilon} \tag{S27}$$

$$u_{\text{nesterov}}^{(t)} = \frac{\beta_1 \hat{m}^{(t)} + (1 - \beta_1)g^{(t)}}{\sqrt{\hat{v}^{(t)}} + \epsilon} \tag{S28}$$

$$\phi^{(t+1)} = \phi^{(t)} - (1 - b_{\text{use nesterov}})\alpha u_{\text{heavy ball}}^{(t)} + \tag{S29}$$

$$b_{\text{use nesterov}}\alpha u_{\text{nesterov}}^{(t)} - \alpha\ell_{2AdamW}\phi^{(t)} \tag{S30}$$

# E    OPTIMIZER FAMILY SEARCH SPACES

## E.1    SEARCH SPACE CONSIDERATIONS

The performance of random search critically depends on the boundaries of the original search space. Without prior knowledge about the problems, however, picking a good search space is difficult. To explore this we additionally choose search spaces *after* collecting and looking at the data. We then use this search space to simulate random search within the constraints via rejection sampling. To find these search spaces we find the best hyper parameters for each task and construct new hyperparameter ranges with min and max values determined by the smallest and largest values of each hyperparameter which were the best hyperparameter for some task. This removes regions of the search space not used by any task. We also tested bounds based on the 5th and 95th percentile of best performing hyperparameters computed over all tasks. In the case of min and max, we find the optimal hyperparameters cover nearly all of the existing space, whereas the percentile based search spaces reduces the volume of the search hypercube by more than 90% leaving us with only ∼100 hyperparameter configurations. In Figure 3, we find, in all cases, learning the hyperparameter list is much more efficient.

## E.2    ADAM8P, ADAM6P, ADAM4P, ADAMLR SEARCH SPACES

For Adam1p, Adam4p, Adam6p, and Adam8p we sample learning rate logritmically between 1e-8 and 10, beta1 and beta2 we parametrize as $1 - x$ and sample logrithmically between 1e-4 and 1 and 1e-6 and 1 respectively. For learning rate schedules we sample linear decay between 1e-7, 1e-4 logrithmically and exponential decay logrithmically between 1e-3, 1e-6. We sample both $\ell_1$ and $\ell_2$ logrithmcally between 1e-8, 1e1.

## E.3    NADAMW SEARCH SPACE

This search space was chosen heuristically in an effort to generalize to new problems. We would like to emphasize that it was not tuned. We used our insight from Adam based optimizer families and

chose this. No iterations where done. We expect more iterations will improve not only in distribution performance, but alsos generalization performance.

The initial learning rate, $\alpha_{base}$ is sampled from log space between $1e-5$ and $1.0$. $1-\beta_1$ is sampled logrithmically between $1e-3$, and $1.0$. $1-\beta_2$ is sampled between $1e-5$, and $1.0$. $\epsilon$ is sampled logarithmically between $1e-8$ and $1e4$. We sample using nesterov ($b_{\text{use nesterov}}$) 50% of the time. We sample $\ell_{2WD}$ and $\ell_{2AdamW}$ logrithmically between $1e-5$ and $1e-1$. Equal probabilities of a third we either use both terms, zero out $\ell_{2WD}$, or zero out $\ell_{2AdamW}$. With 50% probability we use a nonzero min learning rate multiplier sampled logrithmically between $1e-5$ and $1.0$. With 50% probability we sample the warm up fraction, $c_{\text{warmup}}$ between 1e-5 and 1e-1, otherwise it is set to zero. Finally, we uniformly sample the amount of time the learning rate is held constant($c_{\text{constant}}$) between 0 and 1.

## F  EXTENDED RELATED WORK

### F.1  SETS OF TASKS

Benchmarks consisting of multiple tasks are becoming an increasingly common technique for measuring improvement in algorithm design. Reinforcement learning has Atari Bellemare et al. (2013), DMLab Beattie et al. (2016), gym Brockman et al. (2016), and dm_control Tassa et al. (2018). Natural language processing has evaluation sets such as GLUE (Wang et al., 2018), Super GLUE (Wang et al., 2019), and the NLPDecathalon (McCann et al., 2018). In computer vision there is (Zhai et al., 2019) which studies transfer learning of image features. In black box optimization there is Nevergrad (Rapin & Teytaud, 2018), COmparing Continuous Optimizers (COCO) (Hansen et al., 2016) and a number of tasks to test Bayesian hyperparameter optimization presented in (Dewancker et al., 2016). For first order gradient methods there are unit tests for stochastic optimization (Schaul et al., 2013) which studies toy optimization functions, and DeepObs (Schneider et al., 2019) which includes 20 neural network tasks. Hyperparameter tuning practices on these benchmarks vary between tuning on each task separately, to tuning one set of hyperparameters for all problems. In Atari (Bellemare et al., 2013), for example, it is common practice to tune hyperparameters on a subset of tasks and evaluate on the full set. This protocol can further be extended by leveraging unseen levels or games at test time as done in Obstacle Tower (Juliani et al., 2019), ProcGen (Cobbe et al., 2019), CoinRun (Cobbe et al., 2018), and Sonic (Nichol et al., 2018). We believe generalization to unseen tasks is key for learned algorithms to be useful thus our learned search space experiments mirror this setting by making use of hold out tasks.

Existing meta-learning data sets share similar goals to our work but focus on different domains. In few shot learning there is MiniImageNet (Vinyals et al., 2016) which is built procedurally from the ImageNet dataset (Russakovsky et al., 2015). Meta-Dataset (Triantafillou et al., 2019) takes this further and also focuses on generalization by constructing few shot learning tasks using images from a number of different domains for evaluation purposes. The automated machine learning community has OpenML (Vanschoren et al., 2013) with a focus on selecting and tuning non-neural algorithms. For learning optimizers, the use of task suites has been limited and ad-hoc. Many works use a single or small number of standard machine learning tasks (Andrychowicz et al., 2016; Li & Malik, 2017; Lv et al., 2017; Metz et al., 2019a). Wichrowska et al. (2017) uses a set of synthetic problems meant to emulate many different kinds of loss surfaces. While existing collections of tasks exist for optimizer evaluation, e.g. (Schneider et al., 2019), they contain too small a number of tasks to act as a comprehensive training set for learning algorithms, and many of their tasks are additionally too computationally expensive to be useful during learning.

### F.2  HAND DESIGNED AND LEARNED OPTIMIZERS

Optimization is core to machine learning and thus the focus of extensive work. Methods such as Nesterov momentum (Nesterov, 1983), AdaGrad (Duchi et al., 2011), RMSProp (Tieleman & Hinton, 2012), and Adam (Kingma & Ba, 2014) have all shown considerable improvements in both the speed of optimization and ease of use by exposing robust, and easier to tune hyperparameters than SGD (Sivaprasad et al., 2019). Adaptive step size methods in particular have emerged at the forefront with

many works building from it including AdamW (Loshchilov & Hutter, 2017), RAdam (Liu et al., 2019), Novograd (Ginsburg et al., 2019), and NAdam Dozat (2016). Recently, there has been a focus on comparing optimizers either for best performance, or ease of use (Wilson et al., 2017; Choi et al., 2019; Schneider et al., 2019; Sivaprasad et al., 2019). This has proven difficult as performance is heavily dependent on the choice of search space for optimization hyperparameters (Choi et al., 2019).

Learned optimizers represent a parallel thread in the development of optimizers. By learning as opposed to hand-designing optimizers, researchers hope to not only increase performance but also ease of use (e.g. minimize the number of hyperparameters required or lower hyperparameter sensitivity) (Bengio et al., 1990; Schmidhuber, 1995; Hochreiter et al., 2001). Recently, there has been renewed interest in parameterizing learning algorithms with neural networks and learning these optimizers on neural network based losses (Andrychowicz et al., 2016; Wichrowska et al., 2017; Li & Malik, 2017; Lv et al., 2017; Metz et al., 2019a;b). Other approaches make learn symbolic parameterizations for new optimizers (Bello et al., 2017). These various methods are all trained and evaluated on different distributions of tasks making comparison across papers challenging. The dataset of tasks presented here will hopefully aid in the ability to compare and evaluate progress in learned optimizer research.

In this work, we develop a much more minimal type of "learned optimizer" than previous work which developed new functional forms for the optimizer. Optimization involves not only the functional form of the optimizer, but also the rules for choosing hyperparameters and applying the optimizer. We focus on this second aspect of optimization and learn a hyperparameter search space to improve the performance of existing hand designed methods.

### F.3 Hyperparameter search

Hyperparameter search is a key component in machine learning. Considerable improvements have been made in language Melis et al. (2017), computer vision (Snoek et al., 2012), and RL (Chen et al., 2018) simply by tuning better. Often no single hyperparameter configuration works well across all tasks for existing optimization methods. Most current hyperparameter search methods involve trying a very large number of hyperparameters for every new task, which is computationally infeasible for large tasks, and additionally can severely limit the number of hyperparameters that can be tuned. Many common techniques such as random search (Bergstra & Bengio, 2012; Bousquet et al., 2017), Bayesian optimization (Snoek et al., 2012; 2015), tree parzen estimators (Bergstra et al., 2011), or sequential halving (Kumar et al., 2018) require setting a hyperparameter search space by hand which is not only difficult but often wildly inefficient.

Learning hyperparameters or search strategies by leveraging multiple tasks has been explored within the context of Bayesian optimization Swersky et al. (2013); Perrone & Shen (2019); Perrone et al. (2018) as well as under the term meta-learning in Chen et al. (2017) in which an LSTM is meta-trained to produce function locations to query.

The cost of hyperparameter search is often large as each evaluation requires training a model to completion. Often multi-fidelity based approaches are used which leverage "simpler" tasks and transfer the resulting hyperparameters (Hutter et al., 2018). Common approaches include training on partial function evaluations Swersky et al. (2014); Domhan et al. (2015); Li et al. (2016); Klein et al. (2016); Falkner et al. (2018), or leveraging simplified data and models (Petrak, 2000; Zoph & Le, 2016; Brock et al., 2017). Our dataset of tasks serves as a: "simpler" set of tasks to train on; a large and diverse enough set of problems that optimization algorithms trained on it may be expected to generalize; and a framework to test transfer across different types of problems.

# G  LIST OF NADAM HPARAMS

| Idx | Lr | warmup | constant | Min LR mult | beta1 | beta2 | epsilon | nesterov | l2 reg | l2 weight decay |
|-----|------|--------|----------|-------------|---------|---------|-----------|----------|----------|-----------------|
| 0 | 1.24e-3 | 0.000 | 0.477 | 1.01e-3 | 0.94666 | 0.94067 | 8.114e-8 | False | 0.000e+00 | 7.258e-5 |
| 1 | 5.33e-3 | 0.000 | 0.172 | 0.0 | 0.96047 | 0.99922 | 8.665e-8 | True | 0.000e+00 | 5.563e-3 |
| 2 | 2.12e-4 | 0.000 | 0.210 | 1.39e-3 | 0.62297 | 0.97278 | 1.540e-7 | False | 0.000e+00 | 5.361e-2 |
| 3 | 4.06e-1 | 0.000 | 0.324 | 0.0 | 0.99724 | 0.98680 | 1.079e+02 | True | 0.000e+00 | 1.562e-2 |
| 4 | 2.05e-2 | 0.000 | 0.885 | 1.57e-5 | 0.35731 | 0.86043 | 8.874e-5 | True | 0.000e+00 | 7.217e-2 |
| 5 | 5.95e-4 | 0.008 | 0.378 | 0.0 | 0.89130 | 0.99983 | 1.483e-7 | True | 0.000e+00 | 4.087e-2 |
| 6 | 7.53e-3 | 0.000 | 0.422 | 9.55e-4 | 0.69192 | 0.98434 | 3.593e-8 | False | 0.000e+00 | 3.060e-4 |
| 7 | 4.69e-3 | 0.000 | 0.509 | 0.0 | 0.99639 | 0.98820 | 2.056e-5 | False | 0.000e+00 | 3.552e-2 |
| 8 | 2.95e-1 | 0.000 | 0.201 | 0.0 | 0.99678 | 0.99981 | 7.498e+00 | False | 3.792e-4 | 3.463e-4 |
| 9 | 2.04e-3 | 0.000 | 0.527 | 0.0 | 0.49995 | 0.99755 | 5.630e-8 | True | 0.000e+00 | 2.796e-2 |
| 10 | 7.39e-1 | 0.001 | 0.556 | 3.31e-3 | 0.99691 | 0.80639 | 2.900e+03 | False | 0.000e+00 | 7.851e-2 |
| 11 | 8.12e-3 | 0.000 | 0.207 | 0.0 | 0.17785 | 0.96033 | 7.971e-2 | False | 0.000e+00 | 1.489e-2 |
| 12 | 3.33e-2 | 0.000 | 0.369 | 0.0 | 0.69592 | 0.99997 | 5.510e-6 | True | 0.000e+00 | 1.362e-5 |
| 13 | 6.95e-3 | 0.000 | 0.014 | 0.0 | 0.99412 | 0.99305 | 4.352e-7 | False | 0.000e+00 | 3.142e-5 |
| 14 | 1.88e-1 | 0.000 | 0.205 | 1.08e-1 | 0.98597 | 0.56531 | 3.335e+00 | True | 1.265e-5 | 3.868e-3 |
| 15 | 9.47e-4 | 0.007 | 0.452 | 0.0 | 0.43977 | 0.09422 | 2.120e-7 | False | 0.000e+00 | 6.902e-3 |
| 16 | 3.75e-3 | 0.000 | 0.184 | 0.0 | 0.87756 | 0.96128 | 3.163e-3 | True | 7.468e-5 | 2.627e-3 |
| 17 | 7.25e-1 | 0.000 | 0.495 | 0.0 | 0.99800 | 0.99781 | 3.608e+00 | True | 1.656e-5 | 3.911e-2 |
| 18 | 4.58e-3 | 0.000 | 0.107 | 3.66e-1 | 0.42294 | 0.99963 | 4.174e-6 | True | 0.000e+00 | 4.446e-3 |
| 19 | 3.07e-4 | 0.007 | 0.518 | 0.0 | 0.57863 | 0.99625 | 9.881e-6 | False | 0.000e+00 | 5.521e-2 |
| 20 | 2.94e-5 | 0.000 | 0.830 | 8.27e-5 | 0.96916 | 0.99896 | 7.782e-7 | True | 3.364e-4 | 3.416e-3 |
| 21 | 1.65e-4 | 0.002 | 0.457 | 2.70e-1 | 0.95280 | 0.04565 | 2.832e-6 | True | 0.000e+00 | 1.141e-2 |
| 22 | 9.17e-1 | 0.010 | 0.897 | 2.67e-2 | 0.45061 | 0.99244 | 4.945e-1 | False | 1.253e-3 | 0.000e+00 |
| 23 | 2.36e-3 | 0.000 | 0.986 | 0.0 | 0.98560 | 0.99997 | 1.080e-8 | True | 0.000e+00 | 3.023e-3 |
| 24 | 2.14e-2 | 0.000 | 0.128 | 0.0 | 0.98741 | 0.99336 | 1.266e-4 | False | 0.000e+00 | 5.194e-4 |
| 25 | 5.91e-2 | 0.000 | 0.062 | 0.0 | 0.99794 | 0.99383 | 3.447e+02 | True | 0.000e+00 | 3.935e-2 |
| 26 | 1.57e-3 | 0.000 | 0.251 | 0.0 | 0.91820 | 0.99991 | 4.675e-5 | False | 0.000e+00 | 4.112e-5 |
| 27 | 4.43e-1 | 0.000 | 0.702 | 0.0 | 0.94375 | 0.93551 | 2.335e-8 | True | 0.000e+00 | 8.325e-5 |
| 28 | 2.98e-3 | 0.008 | 0.046 | 0.0 | 0.68612 | 0.94232 | 6.614e-2 | False | 6.489e-5 | 0.000e+00 |
| 29 | 1.65e-2 | 0.004 | 0.082 | 4.92e-4 | 0.95717 | 0.99789 | 3.068e+01 | True | 0.000e+00 | 8.920e-2 |
| 30 | 5.58e-3 | 0.000 | 0.538 | 0.0 | 0.97559 | 0.99990 | 3.238e-8 | True | 0.000e+00 | 4.896e-4 |
| 31 | 8.54e-1 | 0.000 | 0.229 | 0.0 | 0.93129 | 0.50200 | 2.051e-2 | False | 2.068e-4 | 2.801e-2 |
| 32 | 7.38e-3 | 0.000 | 0.722 | 8.78e-2 | 0.21456 | 0.99752 | 2.862e-2 | False | 0.000e+00 | 8.439e-2 |
| 33 | 4.26e-4 | 0.001 | 0.923 | 2.06e-1 | 0.47239 | 0.99974 | 8.221e-5 | False | 1.248e-5 | 0.000e+00 |
| 34 | 6.04e-3 | 0.000 | 0.698 | 0.0 | 0.97849 | 0.91449 | 1.806e+00 | False | 3.183e-3 | 1.762e-2 |
| 35 | 8.86e-3 | 0.000 | 0.104 | 1.66e-1 | 0.98967 | 0.99720 | 1.493e-2 | True | 0.000e+00 | 2.253e-2 |
| 36 | 1.51e-2 | 0.000 | 0.431 | 1.99e-3 | 0.80488 | 0.97878 | 2.538e-8 | True | 0.000e+00 | 2.269e-5 |
| 37 | 2.50e-3 | 0.000 | 0.009 | 0.0 | 0.98127 | 0.99988 | 1.799e-7 | False | 0.000e+00 | 1.303e-2 |
| 38 | 3.42e-4 | 0.000 | 0.827 | 6.38e-1 | 0.25217 | 0.96572 | 2.928e-7 | True | 0.000e+00 | 1.318e-3 |
| 39 | 6.94e-5 | 0.000 | 0.085 | 0.0 | 0.98674 | 0.42709 | 2.387e-7 | False | 0.000e+00 | 2.071e-4 |
| 40 | 3.03e-2 | 0.001 | 0.313 | 0.0 | 0.90610 | 0.99997 | 4.449e-3 | True | 0.000e+00 | 2.813e-5 |
| 41 | 4.64e-3 | 0.000 | 0.495 | 2.26e-5 | 0.64658 | 0.54108 | 3.528e-8 | False | 0.000e+00 | 2.996e-5 |
| 42 | 2.25e-3 | 0.000 | 0.722 | 0.0 | 0.97967 | 0.97518 | 1.488e-7 | True | 1.812e-5 | 2.180e-2 |
| 43 | 6.66e-4 | 0.000 | 0.632 | 2.79e-5 | 0.65968 | 0.99997 | 6.848e-6 | True | 0.000e+00 | 3.130e-3 |
| 44 | 3.31e-3 | 0.000 | 0.146 | 0.0 | 0.90447 | 0.99970 | 6.618e-6 | True | 0.000e+00 | 2.184e-2 |
| 45 | 7.84e-4 | 0.016 | 0.124 | 0.0 | 0.95065 | 0.99685 | 2.141e-2 | False | 0.000e+00 | 4.024e-5 |
| 46 | 6.16e-3 | 0.016 | 0.623 | 0.0 | 0.98823 | 0.98744 | 1.616e-6 | False | 0.000e+00 | 1.544e-2 |
| 47 | 3.26e-4 | 0.000 | 0.738 | 1.61e-4 | 0.78425 | 0.99998 | 3.468e-3 | False | 0.000e+00 | 4.709e-2 |
| 48 | 4.12e-3 | 0.001 | 0.205 | 0.0 | 0.99561 | 0.75382 | 2.390e-6 | True | 0.000e+00 | 3.631e-2 |
| 49 | 6.26e-1 | 0.000 | 0.932 | 2.52e-3 | 0.99401 | 0.83521 | 2.431e+00 | True | 0.000e+00 | 1.048e-2 |

Top 50 hyper parameters found using the NAdamW search space. We find diverse learning rates, with very little warmup used. We additionally find most good performing optimizers make use of AdamW style weight decay. Finally, matching insight from (Choi et al., 2019), we find large values of $\epsilon$.

## H    DESCRIPTION OF TASKS IN TASK SUITE

In this section we detail the task distribution used throughout this work. In addition to this text, a Tensorflow (Abadi et al., 2016) implementation is also released at github.com/google-research/google-research/tree/master/task_set.

### H.1    SAMPLED TASKS

#### H.1.1    DEFAULT SAMPLED COMPONENTS

As many of the sampled tasks are neural networks. We define common sampling routines used by all the sampled tasks.

**Activation functions:** We define a distribution of activation functions which is sampled corresponding the following listing both name and weight. These are a mix of standard functions (relu, tanh) to less standard (cos).

- relu: 6
- tanh: 3
- cos: 1
- elu: 1
- sigmoid: 1
- swish (Ramachandran et al., 2017): 1
- leaky relu (with $\alpha = 0.4$): 1
- leaky relu (with $\alpha = 0.2$): 1
- leaky relu (with $\alpha = 0.1$): 1

**Initializations:** We sample initializers according to a weighted distribution. Each initialization sample also optionally samples hyperparameters (e.g. for random normal initializers we sample standard deviation of the underlying distribution).

- he normal (He et al., 2015): 2
- he uniform (He et al., 2015): 2
- glorot normal (Glorot & Bengio, 2010): 2
- glorot uniform (Glorot & Bengio, 2010): 2
- orthogonal: 1. We sample the "gain", or multiplication of the orthogonal matrix logarithmically between $[0.1, 10]$.
- random uniform 1.0: This is defined between $[-s, s]$ where $s$ is sampled logarithmically between $[0.1, 10]$.
- random normal: 1.0: The std is sampled logarithmically between $(0.1, 10)$.
- truncated normal: 1.0: The std is sampled logarithmically between $(0.1, 10)$.
- variance scaling: 1.0: The scale is sampled logarithmically between $(0.1, 10)$.

**RNN Cores:** We define a distribution over different types of RNN cores used by the sequential tasks. With equal probability we sample either a vanilla RNN (Elman, 1990), GRU(Chung et al., 2014), or LSTM(Hochreiter & Schmidhuber, 1997). For each cell we either sample 1 shared initialization method or sample a different initialization method per parameter vector with a 4:1 ratio. We sample the core hidden dimension logarithmically between $[32, 128]$.

#### H.1.2    SAMPLED DATASETS

**Image Datasets:** We sample uniformly from the following image datasets. Each dataset additionally has sampled parameters. For all datasets we make use of four data splits: train, valid-inner, valid-outer, test. Train is used to train models, valid-inner is used while training models to allow for modification

of the training procedure (e.g. if validation loss doesn't increase, drop learning rate). Valid-outer is used to select meta-parameters. Test should not be used during meta-training.

For all datasets, we sample a switch with low probability (10% of the time) to only use training data and thus not test generalization. This ensures that our learned optimizers are capable of optimizing a loss as opposed to a mix of optimizing and generalizing.

**Mnist:** Batch size is sampled logarithmically between $[8, 512]$. We sample the number of training images logarithmically between $[1000, 55000]$ (LeCun, 1998).

**Fashion Mnist:** Batch size is sampled logarithmically between $[8, 512]$. We sample the number of training images logarithmically between $[1000, 55000]$ (Xiao et al., 2017).

**Cifar10:** Batch size is sampled logarithmically between $[8, 256]$. The number of training examples is sampled logarithmically $[1000, 50000]$ (Krizhevsky et al., 2009).

**Cifar100:** Batch size is sampled logarithmically between $[8, 256]$. The number of training examples is sampled logarithmically $[1000, 50000]$ (Krizhevsky et al., 2009).

**{food101_32x32, coil100_32x32, deep_weeds_32x32, sun397_32x32}**: These dataset take the original set of images and resize them to 32x32 using OpenCV's (Bradski, 2000) cubic interpolation. We ignore aspect ratio for this resize. Batch size is sampled logarithmically between $[8, 256]$ (Bossard et al., 2014; Nene et al., 1996; Olsen et al., 2019; Xiao et al., 2010).

**Imagenet32x32 / Imagenet16x16:** The ImageNet 32x32 and 16x16 dataset as created by Chrabaszcz et al. (2017). Batch size is logrithmically sampled between $[8, 256]$.

### H.1.3    TEXT CLASSIFICATION:

**IMDB sentiment classification:** We use text from the IMDB movie reviews dataset(Maas et al., 2011) and tokenize using subwords using a vocab size of 8k(Sennrich et al., 2015). We then take length s random slice from each example where s is sampled logarithmically between $[8, 64]$. These examples are then batched into a batch size logarithmically sampled between $[8, 512]$. We sample the number of training examples logarithmically between $[1000, 55000]$ and with 10% probability just use training data instead of valid / test to test pure optimization as opposed to generalization.

### H.1.4    CHARACTER AND WORD LANGUAGE MODELING

For the character and word language modeling datasets we make use of the following data sources: **imdb movie reviews**(Maas et al., 2011), **amazon product reviews** (ama) using the Books, Camera, Home, and Video subset each as separate datasets, LM1B(Chelba et al., 2013), and **Wikipedia**(Foundation) taken from the 20190301 dump using the zh, ru, ja, hab, and en language codes. We split each article by new lines and only keep resulting examples that contain more than 5 characters. For infrastructure reasons, we only use a million articles from each language and only 200k examples to build the tokenizer.

**Byte encoding:** We take length s random slices of each example where $s$ is sampled logarithmically between $[10, 160]$. These examples are then batched into a batch size logarithmically sampled between $[8, 512]$. With probability 0.2 we restrict the number of training examples to a number logarithmically sampled between $[1000, 50000]$. Finally, with a 10% probability just use training data instead of valid / test to test pure optimization as opposed to generalization.

**subword encoding:** We encode the text as subwords with a vocabsize of 8k (Sennrich et al., 2015). We then take length $s$ random slices of each example where s is sampled logarithmically between $[10, 256]$. These examples are then batched into a batch size logarithmically sampled between $[8, 512]$. With probability 0.2 we restrict the number of training examples to a number logarithmically sampled between $[1000, 50000]$. Finally, with a 10% probability just use training data instead of valid / test to test pure optimization as opposed to generalization.

## H.2 SAMPLED TASKS

### H.2.1 MLP

This task family consists of a multi layer perceptron trained on flattened image data. The amount of layers is sampled uniformly from $[1, 6]$. Layer hidden unit sizes are sampled logarithmically between $[16, 128]$ with different number of hidden units per layer. One activation function is chosen for the whole network and is chosen as described in H.1.1. One shared initializer strategy is also sampled. The image dataset used is also sampled.

Two sampled configurations are shown below.

```
1  {
2    "layer_sizes": [
3      71
4    ],
5    "activation": "leaky_relu2",
6    "w_init": [
7      "he_normal",
8      null
9    ],
10   "dataset": [
11     "sun397_32x32",
12     {
13       "bs": 32,
14       "just_train": false,
15       "num_train": null
16     },
17     {
18       "crop_amount": 0,
19       "flip_left_right": false,
20       "flip_up_down": true,
21       "do_color_aug": false,
22       "brightness": 0.0029364891211851211,
23       "saturation": 0.4308521744067503,
24       "hue": 0.19648945965587863,
25       "contrast": 0.036096320130911644
26     }
27   ],
28   "center_data": false
29 }
```

```
1  {
2    "layer_sizes": [
3      68,
4      37,
5      78
6    ],
7    "activation": "relu",
8    "w_init": [
9      "glorot_normal",
10     null
11   ],
12   "dataset": [
13     "food101_32x32",
14     {
15       "bs": 117,
16       "just_train": true,
17       "num_train": null
18     },
19     null
20   ],
```

```
21     "center_data": true
22   }
```

### H.2.2    MLP_AE

This task family consists of a multi layer perceptron trained with an auto encoding loss. The amount of layers is sampled uniformly from $[2, 7]$. Layer hidden unit sizes are sampled logarithmically between $[16, 128]$ with different number of hidden units per layer. The last layer always maps back to the input dimension. The output activation function is sampled with the following weights: tanh:2, sigmoid:1, linear_center:1, linear:1 where linear_center is an identity mapping. When using the linear_center and tanh activation we shift the ground truth image to $[-1, 1]$ before performing a comparison to the model's predictions. We sample the per dimension distance function used to compute loss with weights l2:2, l1:1, and the reduction function across dimensions to be either mean or sum with equal probability. A single activation function, and initializer is sampled. We train on image datasets which are also sampled.

A sample configurations is shown below.

```
1  {
2    "hidden_units": [
3       73,
4       103,
5       105,
6       104,
7       76
8    ],
9    "activation": "relu",
10   "w_init": [
11     "glorot_uniform",
12     null
13   ],
14   "dataset": [
15     "mnist",
16     {
17        "bs": 39,
18        "num_train": 43753,
19        "num_classes": 10,
20        "just_train": false
21     },
22     null
23   ],
24   "output_type": "tanh",
25   "loss_type": "l2",
26   "reduction_type": "reduce_sum"
27  }
```

### H.2.3    MLP VAE

This task has an encoder with sampled number of layers between $[1, 3]$. For each layer we sample the number of hidden units logarithmically between $[32, 128]$. For the decoder we sample the number of layers uniformly between $[1, 3]$. For each layer we sample the number of hidden units logarithmically between $[32, 128]$. We use a gaussian prior of dimensionality logarithmically sampled between $[32, 128]$. A single activation function and initialization is chosen for the whole network. The output of the encoder is projected to both a mean, and a log standard deviation which parameterizes the variational distribution, $q(z|x)$. The decoder maps samples from the latent space to a quantized gaussian distribution in which we compute data log likelihoods $\log p(x|z)$. The loss we optimize is the evidence lower bound (ELBO) which is computed by adding this likelihood to the kl divergence between our normal distribution prior and $q(z|x)$. We use the reparameterization trick to compute gradients. This model is trained on sampled image datasets.

A sample configuration is listsed below.

```
1  {
2    "enc_hidden_units": [
3      73
4    ],
5    "dec_hidden_units": [
6      74
7    ],
8    "activation": "relu",
9    "w_init": [
10     "he_normal",
11     null
12   ],
13   "dataset": [
14     "food101_32x32",
15     {
16       "bs": 22,
17       "just_train": true,
18       "num_train": null
19     },
20     null
21   ]
22 }
```

### H.2.4 CONV POOLING

This task consists of small convolutional neural networks with pooling. We sample the number of layers uniformly between $[1, 5]$. We sample a stride pattern to be either all stride 2, repeating the stride pattern of 1,2,1,2... for the total number of layers, or 2,1,2,1... for the total number of layers. The hidden units are logarithmically sampled for each layer between $[8, 64]$. We sample one activation function and weight init for the entire network. Padding for the convolutions are sampled per layer to either be same or valid with equal probability. For the convnet we also sample whether or not to use a bias with equal probability. At the last layer of the convnet we do a reduction spatially using either the mean, max, or squared mean sampled uniformly. This reduced output is fed into a linear layer and a softmax cross entropy loss. These models are trained on a sampled image dataset.

A sample configuration is shown below.

```
1  {
2    "strides": [
3      [1, 1],
4      [2, 2],
5      [1, 1],
6      [2, 2],
7      [1, 1]
8    ],
9    "hidden_units": [
10     46,
11     48,
12     47,
13     29,
14     18
15   ],
16   "activation": "leaky_relu4",
17   "w_init": [
18     "glorot_normal",
19     null
20   ],
21   "padding": [
22     "SAME",
```

```
23        "SAME",
24        "VALID",
25        "SAME",
26        "VALID"
27      ],
28      "pool_type": "squared_mean",
29      "use_bias": true,
30      "dataset": [
31        "cifar100",
32        {
33          "bs": 10,
34          "num_train": 5269,
35          "just_train": true
36        },
37        null
38      ],
39      "center_data": false
40    }
```

### H.2.5 CONV FC

This task consists of small convolutional neural networks, flattened, then run through a MLP. We sample the number of conv layers uniformly between $[1, 5]$. We sample a stride pattern to be either all stride 2, repeating the stride pattern of 1,2,1,2... for the total number of layers, or 2,1,2,1... for the total number of layers. The hidden units are logarithmically sampled for each layer between $[8, 64]$. Padding for the convolutions are sampled per layer to either be same or valid with equal probability.

The output is then flattened, and run through a MLP with hidden layers sampled uniformly from $[0, 4]$ and with sizes sampled logrithmically from $[32, 128]$. The loss is then computed via softmax cross entropy.

We sample one activation function and weight init for the entire network. For the convnet we also sample whether or not to use a bias with equal probability. These models are trained on a sampled image dataset.

An example configuration is shown below.

```
1   {
2     "strides": [
3       [2, 2],
4       [2, 2],
5       [2, 2],
6       [2, 2]
7     ],
8     "hidden_units": [
9       17,
10      30,
11      13,
12      16
13    ],
14    "activation": "relu",
15    "w_init": [
16      "glorot_uniform",
17      null
18    ],
19    "padding": [
20      "VALID",
21      "VALID",
22      "VALID",
23      "SAME"
24    ],
```

```
25    "fc_hidden_units": [],
26    "use_bias": true,
27    "dataset": [
28      "coil100_32x32",
29      {
30        "bs": 49,
31        "just_train": false,
32        "num_train": null
33      },
34      null
35    ],
36    "center_data": true
37 }
```

### H.2.6 CHARACTER RNN LANGUAGE MODEL

This task takes character embedded data, and embeds in a size $s$ embedding vector where $s$ is sampled logarithmically between $[8, 128]$ with random normal initializer with std 1.0. With 80% we use all 256 tokens, and with 20% chance we only consider a subset of tokens sampled logarithmically $[100, 256]$. We then pass this embedded vector to a RNN with teacher forcing with equal probability we use a trainable initializer or zeros. A linear projection is then applied to the number of vocab tokens. Losses are computed using a softmax cross entropy vector and mean across the sequence.

A sample configuration is shown below.

```
1  {
2    "embed_dim": 30,
3    "w_init": [
4      "he_normal",
5      null
6    ],
7    "vocab_size": 256,
8    "core": [
9      "gru",
10     {
11       "core_dim": 84,
12       "wh": [
13         "glorot_uniform",
14         null
15       ],
16       "wz": [
17         "random_normal",
18         0.4022641748407826
19       ],
20       "wr": [
21         "he_uniform",
22         null
23       ],
24       "uh": [
25         "he_normal",
26         null
27       ],
28       "uz": [
29         "glorot_normal",
30         null
31       ],
32       "ur": [
33         "glorot_uniform",
34         null
35       ]
36     }
```

```
37      ],
38      "trainable_init": true,
39      "dataset": [
40        "lm1b/bytes",
41        {
42          "patch_length": 147,
43          "batch_size": 63,
44          "just_train": false,
45          "num_train": null
46        }
47      ]
48  }
```

### H.2.7    WORD RNN LANGUAGE MODEL

This task takes word embedded data, and embeds in a size s embedding vector where s is sampled logarithmically between $[8, 128]$ with random normal initializer with std 1.0. A vocab size for this embedding table is sampled logarithmically between $[1000, 30000]$. We then pass this embedded vector to a RNN with teacher forcing with equal probability we use a trainable initializer or zeros. A linear projection is then applied to the number of vocab tokens. Losses are computed using a softmax cross entropy vector and mean across the sequence.

A sample configuration shown below.

```
1   {
2     "embed_dim": 91,
3     "w_init": [
4       "glorot_uniform",
5       null
6     ],
7     "vocab_size": 13494,
8     "core": [
9       "gru",
10      {
11        "core_dim": 96,
12        "wh": [
13          "he_normal",
14          null
15        ],
16        "wz": [
17          "he_normal",
18          null
19        ],
20        "wr": [
21          "he_normal",
22          null
23        ],
24        "uh": [
25          "he_normal",
26          null
27        ],
28        "uz": [
29          "he_normal",
30          null
31        ],
32        "ur": [
33          "he_normal",
34          null
35        ]
36      }
37    ],
```

```
38    "trainable_init": true,
39    "dataset": [
40      "tokenized_amazon_reviews/Video_v1_00_subwords8k",
41      {
42        "patch_length": 14,
43        "batch_size": 59,
44        "just_train": false,
45        "num_train": null
46      }
47    ]
48  }
```

### H.2.8 LOSG PROBLEMS

These tasks consist of a mixture of many other tasks. We sample uniformly over the following types of problems. We brielfy describe them here but refer reader to the provided source for more information. In this work we took all the base problems from (Wichrowska et al., 2017) but modified the sampling distributions to better cover the space as opposed to narrowly sampling particular problem families. Future work will consist of evaluating which sets of problems or which sampling decisions are required.

**quadratic:** n dimensional quadratic problems where n is sampled logarithmically between $[10, 1000]$. Noise is optionally added with probability 0.5 and of the scale s where s is sampled logarithmically between $[0.01, 10]$.

**bowl:** A 2d qaudratic bowl problem with a sampled condition number (logrithmically between $[0.01, 100]$). Noise is optionally added with probability 0.5 and of the scale s where s is sampled logarithmically between $[0.01, 10]$.

**sparse_softmax_regression:** A synthetic random sparse logistic regression task.

**optimization_test_problems:** A uniform sample over the following functions: Ackley, Beale, Branin, logsumexp, Matyas, Michalewicz, Rosenbrock, StyblinskiTang.

**fully_connected:** A sampled random fully connected classification neural network predicting 2 classes on synthetic data. Number of input features is sampled logrithmically between 1 and 16, with a random activation function, and a sampled number of layers uniformly sampled from 2-5.

**norm:** A problem that finds a minimum error in an arbitrary norm. Specifically: $(\sum(Wx - y)^p)^{(\frac{1}{p})}$ where $W \in \mathcal{R}^{NxN}, y \in \mathcal{R}^{Nx1}$. The dimentionality, $N$, is sampled logrithmically between 3, and 1000. The power, $p$, is sampled uniformly between 0.1 and 5.0. $W$, and $y$ are drawn from a standard normal distribution.

**dependency_chain:** A synthetic problem where each parameter must be brought to zero sequentially. We sample dimensionality logrithmically between 3, 100.

**outward_snake:** This loss creates a winding path to infinity. Step size should remain constant across this path. We sample dimensionality logrithmically between 3 and 100.

**min_max_well:** A loss based on the sum of min and max over parameters: $\max x + 1/(\min x) - 2$. Note that the gradient is zero for all but 2 parameters. We sample dimentaionlity logrithmically between 10 and 1000. Noise is optionally added with probability 0.5 and of the scale s where s is sampled logarithmically between $[0.01, 10]$.

**sum_of_quadratics:** A least squares loss of a dimentionality sampled logrithmically between 3 and 100 to a synthetic dataset.

**projection_quadratic:** A quadratic minimized by probing different directions. Dimentionality is sampled from 3 to 100 logrithmically.

In addition to these base tasks, we also provide a variety of transformations described bellow. The use of these transformations is also sampled.

**sparse_problems:** With probability 0.9 to 0.99 the gradient per parameter is set to zero. Additional noise is added with probability 0.5 sampled from a normal with std sampled logrithmically between $[0.01, 10.0]$.

**rescale_problems:** Rescales the loss value by 0.001 to 1000.0 sampled logrithmically.

**log_objective:** Takes the log of the objective value.

2 Sample configurations shown below.

```
[
  "fully_connected",
  {
    "n_features": 16,
    "n_classes": 2,
    "activation": "leaky_relu2",
    "bs": 7,
    "n_samples": 12,
    "hidden_sizes": [
      32,
      8,
      5,
      9,
      8
    ]
  },
  36641
]
```

```
[
  "outward_snake",
  {
    "dim": 9,
    "bs": 30,
    "n_samples": 249
  },
  79416
]
```

```
[
  "rescale_problems",
  {
    "base": [
      "sum_of_quadratics",
      {
        "dim": 36,
        "bs": 5,
        "n_samples": 1498
      }
    ],
    "scale": 227.86715292020605
  },
  89629
]
```

### H.2.9 MASKED AUTOREGRESSIVE FLOWS

Masked autoregressive flows are a family of tractable density generative models. See XX for more information. The MAF is defined by a sequence of bijectors. For one bijector samples a number of layers to either be 1 or 2 with equal probability, and a number of hidden layers sampled logarithmically between $[16, 128]$. We sample the number of bijector uniformly from $[1, 4]$ and use the same hidden

layers across all bijector. We sample activation function, and initializer once for the whole model. In this task we model image datasets which are also sampled.

A sample configuration is shown below.

```
1  {
2    "activation": "relu",
3    "w_init": [
4      "he_uniform",
5      null
6    ],
7    "dataset": [
8      "imagenet_resized/16x16",
9      {
10       "bs": 19,
11       "just_train": true,
12       "num_train": null
13     },
14     null
15   ],
16   "hidden_units": [
17     44,
18     24
19   ],
20   "num_bijectors": 3
21 }
```

### H.2.10  NON VOLUME PRESERVING FLOWS

NVP are a family of tractable density generative models. See Dinh et al. (2016) for more information. The NVP is defined by a sequence of bijectors. For one bijector samples a number of layers to either be 1 or 2 with equal probability, and a number of hidden layers sampled logarithmically between $[16, 128]$. We sample the number of bijector uniformly from $[1, 4]$ and use the same hidden layers across all bijector. We sample activation function, and initializer once for the whole model. In this task we model image datasets which are also sampled.

A sample configuration shown below.

```
1  {
2    "activation": "cos",
3    "w_init": [
4      "glorot_normal",
5      null
6    ],
7    "dataset": [
8      "sun397_32x32",
9      {
10       "bs": 228,
11       "just_train": false,
12       "num_train": null
13     },
14     null
15   ],
16   "hidden_units": [
17     21,
18     121
19   ],
20   "num_bijectors": 4
21 }
```

### H.2.11 QUADRATIC LIKE PROBLEMS

This task distribution defines a synthetic problem based on a non-linear modification to a quadratic. The dimensionality of the problem is sampled logarithmically between [2, 3000].

The loss for this task is described by:

$$\text{output\_fn}((AX - B)^2 + C) \tag{S31}$$

where $X = \text{param} * \text{weight\_rescale}$ and where param is initialized by initial_dist.sample() / weight_rescale.

The output_fn is sampled uniformly between identity, and $f(x) = \log(\max(0, x))$. The loss scale is sampled logarithmically between $[10^{-5}, 10^3]$.

We define a distribution over matrices A as a sample from one of the following: normal: we sample a mean from a normal draw with a standard deviation of 0.05 and a std from a uniform [0, 0.05]. The elements of A are drawn from the resulting distribution. uniform: linspace_eigen: logspace_eigen:

We define a distribution over B to be either normal with mean and std sampled from N(0, 1), U(0, 2) respectively or uniform with min and range equal to U(-5, 2.5), U(0, 5) respectively.

With probability 50% we add noise from a distribution whose parameters are also sampled.

A sample configuration shown below.

```
{
  "A_dist": [
    "linspace_eigen",
    {
      "min": 32.09618575514275,
      "max": 122.78045861480965
    }
  ],
  "initial_dist": [
    "uniform",
    {
      "min": 2.3911997838130956,
      "max": 6.723940057771417
    }
  ],
  "output_fn": "log",
  "dims": 212,
  "seed": 68914,
  "loss_scale": 0.6030061302850566,
  "noise": null
}
```

### H.2.12 RNN TEXT CLASSIFICATION

This task consists of using an RNN to classify tokenized text. We first trim the vocab length to be of a size logarithmically sampled between [100, 10000]. The text is then embedded into a vocab size logarithmically sampled between [8, 128]. These embeddings get fed into a sampled config RNN. With equal probability the initial state of the rnn is either sampled, or zeros. With equal probability we either take the last RNN prediction, the mean over features, or the per feature max over the sequence. This batch of activations is then passed through a linear layer and a softmax cross entropy loss. The initialization for the linear projection is sampled.

An example configuration shown below. In this version of TaskSet the dataset sampling contains a bug. All data used is from the imdb_reviews/subwords8k dataset.

```
{
  "embed_dim": 111,
```

```
 3    "w_init": [
 4      "random_normal",
 5      0.1193048629073732
 6    ],
 7    "dataset": [
 8      "imdb_reviews/subwords8kimdb_reviews/bytes",
 9      {
10        "bs": 43,
11        "num_train": null,
12        "max_token": 8185,
13        "just_train": true,
14        "patch_length": 20
15      }
16    ],
17    "vocab_size": 3570,
18    "core": [
19      "vrnn",
20      {
21        "hidden_to_hidden": [
22          "he_uniform",
23          null
24        ],
25        "in_to_hidden": [
26          "he_uniform",
27          null
28        ],
29        "act_fn": "leaky_relu2",
30        "core_dim": 35
31      }
32    ],
33    "trainable_init": false,
34    "loss_compute": "max"
35  }
```

## H.3 FIXED TASKS

In addition to sampled tasks, we also define a set of hand designed and hand specified tasks. These tasks are either more typical of what researcher would do (e.g. using default initializations) or specific architecture features such as bottlenecks in autoencoders, normalization, or dropout.

In total there are 107 fixed tasks. Each task is labeled by name with some information about the underlying task. We list all tasks, discuss groups of tasks, but will not describe each task in detail. Please see the source for exact details.

**Associative_GRU128_BS128_Pairs10_Tokens50**
**Associative_GRU256_BS128_Pairs20_Tokens50**
**Associative_LSTM128_BS128_Pairs10_Tokens50**
**Associative_LSTM128_BS128_Pairs20_Tokens50**
**Associative_LSTM128_BS128_Pairs5_Tokens20**
**Associative_LSTM256_BS128_Pairs20_Tokens50**
**Associative_LSTM256_BS128_Pairs40_Tokens100**
**Associative_VRNN128_BS128_Pairs10_Tokens50**
**Associative_VRNN256_BS128_Pairs20_Tokens50**

These tasks use RNN's to perform an associative memory task. Given a vocab of tokens, and some number of pairs to store and a query the RNN's goal is to produce the desired value. For example given the input sequence A1B2C3?B_ the RNN should produce ________B.

This model embeds tokens, applies an RNN, and applies a linear layer to map back to the output space. Softmax cross entropy loss is used to compare outputs. A weight is also placed on the losses so that loss is incurred only when the RNN is supposed to predict. For RNN cells we use LSTM (Hochreiter & Schmidhuber, 1997), GRU (Chung et al., 2014), and VRNN – a vanilla RNN. The previous tasks are defined with the corresponding RNN cell, number of units, batch size, sequence lengths, and number of possible tokens for the retrieval task.

**Copy_GRU128_BS128_Length20_Tokens10**
**Copy_GRU256_BS128_Length40_Tokens50**
**Copy_LSTM128_BS128_Length20_Tokens10**
**Copy_LSTM128_BS128_Length20_Tokens20**
**Copy_LSTM128_BS128_Length50_Tokens5**
**Copy_LSTM128_BS128_Length5_Tokens10**
**Copy_LSTM256_BS128_Length40_Tokens50**
**Copy_VRNN128_BS128_Length20_Tokens10**
**Copy_VRNN256_BS128_Length40_Tokens50**

These tasks use RNN's to perform a copy task. Given a vocab of tokens and some number of tokens the RNN's job is to read the tokens and to produce the corresponding outputs. For example an input might be: ABBC|____ and the RNN should output ____|ABBC. See the source for a complete description of the task. Each task in this set varies the RNN core, as well as the dataset structure.

This model embeds tokens, applies an RNN, and applies a linear layer to map back to the output space. Softmax crossentropy loss is used to compare outputs. A weight is also placed on the losses so that loss is incurred only when the RNN is supposed to predict. For RNN cells we use LSTM (Hochreiter & Schmidhuber, 1997), GRU (Chung et al., 2014), and VRNN – a vanilla RNN. The previous tasks are defined with the corresponding RNN cell, number of units, batch size, sequence lengths, and number of possible tokens.

**FixedImageConvAE_cifar10_32x32x32x32x32_bs128**
**FixedImageConvAE_cifar10_32x64x8x64x32_bs128**
**FixedImageConvAE_mnist_32x32x32x32x32_bs128**
**FixedImageConvAE_mnist_32x64x32x64x32_bs512**

**FixedImageConvAE_mnist_32x64x8x64x32_bs128**

Convolutional autoencoders trained on different datasets and with different architectures (sizes of hidden units).

**FixedImageConvVAE_cifar10_32x64x128x64x128x64x32_bs128**
**FixedImageConvVAE_cifar10_32x64x128x64x128x64x32_bs512**
**FixedImageConvVAE_cifar10_32x64x128x64x32_bs128**
**FixedImageConvVAE_cifar10_64x128x256x128x256x128x64_bs128**
**FixedImageConvVAE_mnist_32x32x32x32x32_bs128**
**FixedImageConvVAE_mnist_32x64x32x64x32_bs128**
**FixedImageConvVAE_mnist_64x128x128x128x64_bs128**

Convolutional variational autoencoders trained on different datasets, batch sizes, and with different architectures.

**FixedImageConv_cifar100_32x64x128_FC64x32_tanh_variance_scaling_bs64**
**FixedImageConv_cifar100_32x64x64_flatten_bs128**
**FixedImageConv_cifar100_bn_32x64x128x128_bs128**
**FixedImageConv_cifar10_32x64x128_flatten_FC64x32_tanh_he_bs8**
**FixedImageConv_cifar10_32x64x128_flatten_FC64x32_tanh_variance_scaling_bs64**
**FixedImageConv_cifar10_32x64x128_he_bs64**
**FixedImageConv_cifar10_32x64x128_largenormal_bs64**
**FixedImageConv_cifar10_32x64x128_normal_bs64**
**FixedImageConv_cifar10_32x64x128_smallnormal_bs64**
**FixedImageConv_cifar10_32x64x128x128x128_avg_he_bs64**
**FixedImageConv_cifar10_32x64x64_bs128**
**FixedImageConv_cifar10_32x64x64_fc_64_bs128**
**FixedImageConv_cifar10_32x64x64_flatten_bs128**
**FixedImageConv_cifar10_32x64x64_tanh_bs64**
**FixedImageConv_cifar10_batchnorm_32x32x32x64x64_bs128**
**FixedImageConv_cifar10_batchnorm_32x64x64_bs128**
**FixedImageConv_coil10032x32_bn_32x64x128x128_bs128**
**FixedImageConv_colorectalhistology32x32_32x64x64_flatten_bs128**
**FixedImageConv_food10164x64_Conv_32x64x64_flatten_bs64**
**FixedImageConv_food101_batchnorm_32x32x32x64x64_bs128**
**FixedImageConv_mnist_32x64x64_fc_64_bs128**
**FixedImageConv_sun39732x32_bn_32x64x128x128_bs128**
**Mnist_Conv_32x16x64_flatten_FC32_tanh_bs32**

Convolutional neural networks doing supervised classification. These models vary in dataset, architecture, and initializations.

**FixedLM_lm1b_patch128_GRU128_embed64_avg_bs128**
**FixedLM_lm1b_patch128_GRU256_embed64_avg_bs128**
**FixedLM_lm1b_patch128_GRU64_embed64_avg_bs128**
**FixedLM_lm1b_patch128_LSTM128_embed64_avg_bs128**
**FixedLM_lm1b_patch128_LSTM256_embed64_avg_bs128**

Language modeling tasks on different RNN cell types and sizes.

**FixedMAF_cifar10_3layer_bs64**
**FixedMAF_mnist_2layer_bs64**
**FixedMAF_mnist_3layer_thin_bs64**

Masked auto regressive flows models with different architectures (number of layers and sizes).

**FixedMLPAE_cifar10_128x32x128_bs128**
**FixedMLPAE_mnist_128x32x128_bs128**
**FixedMLPAE_mnist_32x32x32_bs128**

Autoencoder models based on multi layer perceptron with different number of hidden layers and dataset.

**FixedMLPVAE_cifar101_128x128x32x128x128_bs128**
**FixedMLPVAE_cifar101_128x32x128_bs128**
**FixedMLPVAE_food10132x32_128x64x32x64x128_bs64**
**FixedMLPVAE_mnist_128x128x8x128_bs128**
**FixedMLPVAE_mnist_128x64x32x64x128_bs64**
**FixedMLPVAE_mnist_128x8x128x128_bs128**
**Imagenet32x30_FC_VAE_128x64x32x64x128_relu_bs256**

Variational autoencoder models built from multi layer perceptron with different datasets, batchsizes, and architectures.

**FixedMLP_cifar10_BatchNorm_128x128x128_relu_bs128**
**FixedMLP_cifar10_BatchNorm_64x64x64x64x64_relu_bs128**
**FixedMLP_cifar10_Dropout02_128x128_relu_bs128**
**FixedMLP_cifar10_Dropout05_128x128_relu_bs128**
**FixedMLP_cifar10_Dropout08_128x128_relu_bs128**
**FixedMLP_cifar10_LayerNorm_128x128x128_relu_bs128**
**FixedMLP_cifar10_LayerNorm_128x128x128_tanh_bs128**
**FixedMLP_cifar10_ce_128x128x128_relu_bs128**
**FixedMLP_cifar10_mse_128x128x128_relu_bs128**
**FixedMLP_food10132x32_ce_128x128x128_relu_bs128**
**FixedMLP_food10132x32_mse_128x128x128_relu_bs128**
**FixedMLP_mnist_ce_128x128x128_relu_bs128**
**FixedMLP_mnist_mse_128x128x128_relu_bs128**
**FixedNVP_mnist_2layer_bs64**

Image classification based on multi layer perceptron. We vary architecture, data, batchsize, normalization techniques, dropout, and loss type across problems.

**FixedNVP_mnist_3layer_thin_bs64**
**FixedNVP_mnist_5layer_bs64**
**FixedNVP_mnist_5layer_thin_bs64**
**FixedNVP_mnist_9layer_thin_bs16**

Non volume preserving flow models with different batchsizesm and architectures.

**FixedTextRNNClassification_imdb_patch128_LSTM128_avg_bs64**
**FixedTextRNNClassification_imdb_patch128_LSTM128_bs64**
**FixedTextRNNClassification_imdb_patch128_LSTM128_embed128_bs64**
**FixedTextRNNClassification_imdb_patch32_GRU128_bs128**
**FixedTextRNNClassification_imdb_patch32_GRU64_avg_bs128**
**FixedTextRNNClassification_imdb_patch32_IRNN64_relu_avg_bs128**
**FixedTextRNNClassification_imdb_patch32_IRNN64_relu_last_bs128**
**FixedTextRNNClassification_imdb_patch32_LSTM128_E128_bs128**
**FixedTextRNNClassification_imdb_patch32_LSTM128_bs128**
**FixedTextRNNClassification_imdb_patch32_VRNN128_tanh_bs128**
**FixedTextRNNClassification_imdb_patch32_VRNN64_relu_avg_bs128**
**FixedTextRNNClassification_imdb_patch32_VRNN64_tanh_avg_bs128**

RNN text classification problems with different RNN cell, sizes, embedding sizes, and batchsize.

**TwoD_Bowl1**
**TwoD_Bowl10**
**TwoD_Bowl100**
**TwoD_Bowl1000**

2D quadratic bowls with different condition numbers.

**TwoD_Rosenbrock**
**TwoD_StyblinskiTang**
**TwoD_Ackley**
**TwoD_Beale**

Toy 2D test functions.

