# OpenReview forum: "TaskSet: A Dataset of Optimization Tasks"
_ICLR.cc/2021/Conference — Reject_

### Official Review · AnonReviewer4 · 2020-10-27
**Collection of tasks**

**Rating:** 3
**Confidence:** 4

**Review:**

This work presents TaskSet, a collection of optimization tasks consisting of different combinations of data, loss function, and network architecture. The tasks are useful when choosing and evaluating different optimizers (e.g. ADAM) for learning tasks. The usefulness of this collection is demonstrated for a hyperparameter search problem.

The main question I had about this work is why is the chosen collection the right set of tasks to be considering? Do I have any assurance that an optimizer chosen using TaskSet will be any good on future tasks? How can we know that we don't overfit to these particular tasks when choosing an optimizer? Is there any notion of two tasks being drawn from the same distribution?

However, my main concern with this paper is that, while TaskSet may be a useful tool for facilitating future research, it is not clear to me that it itself represents an advancement of novel research, which I think should be the bar for acceptance to a major conference. The work does not make any claims, or present any results beyond a use-case for the set of tasks. That's not to say that TaskSet isn't a useful tool, helpful for future research. But it itself does not represent such research. Because of this I recommend the work be rejected.

---

> ### Author Response · Authors · 2020-11-19
> **Response**
>
> Thank you for your thoughtful review.
>
> \> Justification for the selection of tasks:
>
> Much like any benchmark or dataset in machine learning research, we have to agree as a community on the value of a particular benchmark in driving forward progress on problems we care about. For example, image classification datasets such as Cifar10 or MNIST have particularly chosen class labels in particular domains, which only be partially correlated with the overall task of interest (computer vision), yet we have made a ton of progress in computer vision by using these datasets as a standard benchmark for comparing ideas.
>
> Similarly, we chose these particular tasks as they are representative of optimization problems in deep learning. If the reviewer has specific suggestions for problems to include or remove, we are open to discussing them. However, we stand by the general point that there is considerable value in the community adopting and agreeing upon a benchmark, even if all community participants do not necessarily agree on every last detail of every task that makes up the benchmark.
>
> Moreover, we, performed a large number of experiments in the hopes of empirically justifying the choices we made using our learned hyperparameter list. In particular, we explored the number of tasks proposed in figure 3ab where we find that more tasks increasing leads to better performance. With respect to model size, we show in figure S2 that having a wider diversity of model sizes allows our optimizer list to work well across model sizes whereas just training on small tasks does not generalize to larger tasks. In figure S7, we show a justification for having multiple task types, since all of the included task types behave differently during optimization.
>
> \> Does this represent an advancement in research?
>
> We appreciate that you think this will be good for facilitating future research. We disagree, however, that datasets do not represent novel research. Dataset papers are among the most widely used and widely cited contributions to the community. They enable new research directions and applications. They have also been accepted to numerous top tier conferences including, but not limited to: caltech101, imagenet, and Clic from CVPR, SuperGLUE from NeurIPS, LAMBADA, and HellaSwag from ACL, CLEVR from IEEE, META-DATASET, Deepobs, and HolStep from ICLR. In our work we do present a use case of this dataset and show, despite being an incredibly simple method, that the result works well on problems that are of interest to the research community (imagenet+resnets, lm1b+transformers).

---

### Official Review · AnonReviewer2 · 2020-10-28
**Significantly important contribution for developing learned optimizers; justification for task choices and presented application somewhat limited**

**Rating:** 7
**Confidence:** 4

**Review:**

The paper presents a suite of deep learning focused optimization problems that would facilitate the development of learned optimizers. This is very useful and can streamline research in learned optimizers while providing a benchmarking suite that can be used for training as well as evaluation. In my opinion, this is very valuable.

However, the presentation of the task suite and then the subsequent application could use significant clarification. For example, it was not clear to me until I made multiple passes that the application presented in the paper is about meta-learning hyper-parameter initializations for deep learning optimizers by levaraging the suite to generate the meta-learning data set with various task executions. Moreover, given that the main advantage, to the best of my understanding, of the proposed TaskSet is in learning optimizers, the choice of tasks in the paper lack proper qualitative or quantitative justification. As mentioned in the paper, it is useful to have a wide set of tasks for better generalization of learned optimizers, too broad a set could hinder the meta-learning. In that case, it is not clearly discussed (at least in the main paper) why these set of tasks are selected and why/how they lead to better learned optimizers. If the suite consists of all combinations of deep learning architectures, applications and data sets without any meaningful discussion of why it is better for learning optimizers (or even for meta-learning hyper-parameter initializations) than just learning optimizers separately on each task (or even task class), it reduces the possible usefulness of the suite.


Given the motivation and the utility of the suite, I am leaning towards an accept. However, the lack of proper justification for the choices (beyond just covering a laundry list of deep learning architectures, data sets and applications) to the best of my understanding leaves me partially unsatisfied.


Beyond the above, I have a few minor comments:


- Subsection 1.2 is very confusing. Why are we talking about hyperparameters if the taskset consists of first order optimization problems? It should better (and more explicitly) presented as a potential application of the TaskSet suite for meta-learning hyper-parameter initializations for some optimizers.
- Figure 1b is confusing to me. I'm unable to wrap my head around what each line means and how the x axis ordered and why the fraction is always increasing? Moreover, given the nonconvex and randomized nature of many of the discussed tasks, how is it decided that an optimizer achieves a particular loss -- the optimizer might reach different level with different restarts.
- This paper shows an application of TaskSet for meta-learning hyper-parameters of specific optimization algorithms. I am not sure, in the context of HPO, why one would use the TaskSet given something like the Bayesmark framework [A]. It has HPO tasks with associated data with ability to add new tasks/data, and the task set covers a wider class of methods.


[A] bayesmark.readthedocs.io

---

> ### Author Response · Authors · 2020-11-19
> **Response**
>
> Thank you for your thoughtful review.
>
> \> Clarity:
>
> We are sorry to hear you found this aspect confusing and appreciate your feedback on this. We are revising our paper accordingly, and focusing on section 1.2 in particular.
>
> \>Justification for which tasks to include:
>
> As part of our experiments with the learned hyperparameter lists we tried to target sensitivity to particular choices of dataset design. We explored the number of distinct tasks proposed in figure 3ab where we find that training on more tasks leads to better performance. With respect to model size, we show in figure S2 that having a wider diversity of model sizes allows our optimizer list to work well across model sizes whereas only training on small tasks does not generalize to larger tasks. In figure S7, we show a justification for having multiple different kinds of tasks. We find that the tasks are not similar to each other / there are no duplicates. This justifies that each entry of the "laundry list" provides something different.
>
> Additionally, we show the resulting hyperparameter lists transfer. This would not be possible without problems that are at least somewhat aligned to more realistic tasks.
>
> The hope, of course, is to demonstrate that general learned optimizers can be trained to outperform simple per-task hyperparameter tuning of traditional optimizers.  This task set was directly motivated for this purpose, and by the need for a dataset that properly encapsulated the breadth of problems considered in the machine learning literature.  We would argue it is not even really yet known what "tasks [that] lead to better learned optimizers" are, much less how to select them.  However, it was clear that no existing datasets of optimization tasks were nearly large enough to facilitate meaningful transfer or generalization results.  To draw an analogy with Imagenet, it isn't that we think that solving vision [solving learned optimization] requires a class of images involving geological formations [some specific MLP architecture in Taskset], more that we needed a diverse set of tasks to have enough coverage on the types of problems machine learning practitioners care about.
>
>
>
> \>Figure 1b is confusing to me … given the nonconvex and randomized nature of many of the discussed tasks, how is it decided that an optimizer achieves a particular loss -- the optimizer might reach different level with different restarts.
>
> Each point on figure 1b represents training a hyperparameter list using a given number of tasks (x-axis), then applying that list using some number of trials (color / legend). We then report the best performance over these optimizers for each task and present the mean over multiple tasks.
>
> As per the non-convex and randomized nature of the algorithms this is true. We account for this in 2 ways: First, for every result presented we always run >=5 random seed and show averages. Second, when performing max / min operations we always perform an argmax over a subset of the trials, then evaluate the argmax on the remaining trials. This is a technique inspired by double-q learning.
>
>
> \> Relation to Bayesmark:
>
>
> Bayesmark is a library that focuses on comparing HPO methods on tabular data on sklearn models. This is quite a different target than we present here. We focus on deep learning methods, and in this regime we capture much more diversity (it looks like Bayesmark only has MLP's with adam and sgd). Finally, it would be difficult to use bayesmark to train our hyperparameter lists, or other kinds of learned optimizers. Bayesmark instead appears to be used to test an algorithm on a particular task as its primary purpose is benchmarking, whereas taskset can be used for benchmarking but is mainly targeted towards meta-training.

---

### Official Review · AnonReviewer1 · 2020-10-30
**Better comparison with regular hyperparameter search is required**

**Rating:** 5
**Confidence:** 2

**Review:**

This paper proposes a new dataset which contains experiment / model details coupled with optimizer information so as to model the behavior of optimizer, and their effect on performance on test set. The paper is not very difficult to follow, but I am not super convinced of an actual practical use cases.

I think that the authors should provide a concrete examples for real life test time applications. I suppose the meta-learning algorithm for the optimizer would take the experiment definition, and map this information to an optimal optimizer, but I think that it would be easier for the reader if this information could be made mode explicit in the paper, perhaps with a concrete example.

I also think that the comparison between the proposed meta learning approach, vs. the regular hyperparameter search on a given dataset should be made clearer. Right now it is limited to figure 3, and in my opinion the details on how the random search is carried out is not clear enough. What is the range of hyper parameters that are sampled? What are the distributions from which the hyperparameters come from?

Also, it is hard to make the conclusion only from the experiments provided in Figure 3 that, the proposed meta-learning approach would be preferable over the standard hyperparameter search just by the two tasks explored in this particular figure. Ideally a third dimension of tasks should also be added to the figures so that we know that this meta learning approach generalizes over a variety of tasks. (The same comments apply to figure 4, if I understand correctly, which does similar experiments on more realistic models/tasks)

If I am missing something that is already in the paper, I apologize, but without further experimental evidence which suggests that the proposed meta learning scheme would be clearly preferable over standard hyperparameter search, it is hard to see a clear-cut application for this paper.

I appreciate the ambitious task that this paper is trying to tackle, but I feel more convincing experimental evidence, and better presentation of the experiments is required to consolidate the case that this paper is trying to make.

---

> ### Author Response · Authors · 2020-11-19
> **Response**
>
> Thank you for your thoughtful reviewer!
>
> We believe we have addressed your technical concerns, and demonstrated that many of the gaps or extensions you suggest are already present in the paper. We hope you will consider raising your score as a result.
>
> \> I think that the authors should provide a concrete examples for real life test time applications.
>
> This is already done. If we are understanding you correctly, this is exactly what we show in the second half of the paper with the hyperparameter lists. This list treats taskset as a dataset and meta-learns the entries.
>
> \> I also think that the comparison between the proposed meta learning approach, vs. the regular hyperparameter search on a given dataset should be made clearer. Right now it is limited to figure 3 ...
>
> This is already done. We provide 5 figures showing comparisons against random search. In Figure 2 we show random search with multiple different search spaces (1 parameter adam, 4 parameter adam, 8 parameter adam) as well as search spaces with best case bounding boxes (min/max and perc). In figure 3cd we show generalization performance -- I think what you are suggesting with your 3rd axis. We find our hyperparmater list on some set of tasks (e.g. non-RNN), then apply it to a different family of tasks (e.g. RNN tasks). For these figures, these curves are all averages over multiple tasks -- not just one. In Figure 4 we also compare to random search but this time on vastly different tasks.
>
> \> the details on how the random search is carried out is not clear enough. What is the range of hyper parameters that are sampled? What are the distributions from which the hyperparameters come from?
>
> This information is already included. The details of the random search for each experiment are listed in Appendix E.
>
> \>  Ideally a third dimension of tasks should also be added to the figures so that we know that this meta learning approach generalizes over a variety of tasks. (The same comments apply to figure 4, if I understand correctly, which does similar experiments on more realistic models/tasks)
>
> If we understand the suggestion correctly, this analysis of generalization performance to out of distribution tasks is already included in Figures 3c, 3d, and 4, all of which only consider the performance of the resulting OptList on tasks not used for meta-training.
>
> \> Practical use cases
>
> In terms of practical use cases of this paper we provide a few contributions. The first is a dataset and training tasks to enable research, especially in meta-learning of optimizers. The second is an exploration of existing work with a focus on understanding generalization. Finally, and immediately practically useful, is the learned hyper parameter list. We validate this on larger tasks (Figure 4) and believe it provides value -- especially if the researcher doesn't know near optimal hyperparameters to train a target model as is the case for new architectures.
>
> Thank you again for volunteering your time to review our paper.

---

### Official Review · AnonReviewer3 · 2020-10-31
**A dataset of optimization tasks is an interesting contribution, but lacking depth in the discussion of its construction and intended use**

**Rating:** 5
**Confidence:** 4

**Review:**

Summary:

This paper proposes a dataset of tasks to help evaluate learned optimizers. The learned optimizers are evaluated by the loss that they achieve on held-out tasks after 10k steps. Using this dataset, the main strategy considered is to use search spaces that parametrize optimizers and learn a list of hyperparameter configurations for the optimizer that are tried sequentially. The authors show that the learned hyperparameter configuration list learned achieves better performance than (constrained) random search on multiple optimizer search spaces. Finally, they show that the learned hyperparameter list transfer well to realistic problems such as training a ResNet-50 model on ImageNet and training a transformer architecture on LM1B, outperforming reasonable baselines.

Pros:

+ Creating a dataset of tasks for learning optimizers is a interesting and useful goal. While there have been some sets of tasks used in the learned optimizers literature, there isn't a standard dataset for this task. A large number of tasks is proposed.
+ The hyperparameter list trained compares favorably with random search across the other tasks. The experiments are interesting overall and show some insights about the performance of the learned list with increasing number of tasks.

Cons:

- While the goal of finding a good dataset of tasks for learned optimizers is a worthy, I find that the paper does not adequately discuss and explore the choices that went into creating this dataset. Namely, how were tasks picked? What code implementations were used? What are some limitations of the current dataset that could be addressed in future research? How are the tasks represented? How can a researcher use this dataset of tasks to explore new algorithms? Most of the value of proposing a new benchmark or dataset is explaining the choices that went into creating it and packaging well so that other researchers can use it easily. I think that this could be better realized in the paper. For example, is future research based on this dataset meant to be done offline (on the optimization curves collected) or online (by running additional configurations for these search spaces)? How are new methods to be benchmarked through this dataset? How are existing datasets for this missing important aspects? This is not adequately defined. Given this, I think that the paper could do a better job setting the stage for future research building on this dataset.

- The focus on the dataset of tasks is poorly realized in the paper, which is devoted in great part to how an ordered list of 1000 hyperparameter configurations for a learned optimizer performs well in comparison with other random search baselines. While this was well executed overall, it is not initially the focus of the paper. I believe that more value for the community could be derived by focusing on the creation of the dataset rather than on the introduction of a new heuristic.

Typo: . on We take the TPU... ==> . We take the TPU...

---

> ### Author Response · Authors · 2020-11-19
> **Response**
>
> Thank you for your thoughtful review. We are glad you agree that the creation of a dataset of tasks is useful to the community and appreciate your positive view on the execution of the hyperparameter list.
>
> \> Choices that went into the task distribution:
>
> We agree more discussion is needed to this end. In particular, the rationale for which tasks we included. We plan to update the text. In short, our decisions were informed to satisfy the tradeoff between "realistic models" -- model the machine learning / deep learning research community might care about, balanced against the realities of computational cost. We don't know if our exact selection is the "right" distribution of tasks, nor do we think we ever will. We do strive to provide evidence that this distribution does capture properties of real problems which allows us to transfer hparam lists. One big rationale for a large number of the hyperparameter list experiments (fig 3, 4, s2, s3, s4, s7) is to explore different choices in task selection as well as demonstrate alignment to real problems -- providing justification for our selection.
>
> As per how this implementation was done, we provide an in depth appendix as well as open source code written in tensorflow. We will add some more information about this in the main text.
>
> Finally, how this dataset will be used. At this point, we are not sure how this dataset will be used and don't want to artificially impose restrictions. This is a new area of research and imposing a rigid experimental procedure might limit use cases we did not expect. The data we provide (training curves across different baselines) should be useful to researchers in constructing whatever baselines are needed on this dataset.
>
> \> Focus on dataset of tasks is poorly realized -- a lot of paper is a hyperparameter list.
>
> We showed an in depth analysis of the types of research, and questions that could be asked with this dataset as well as used this to probe design choices for TaskSet. Given how non-standard this dataset is, we thought this would be appreciated. We have a large amount more content on the dataset itself in the appendix however (C, E, H) and will shift some of this content to the main body.

---

### Decision · Program_Chairs · 2021-01-07
**Final Decision**

**Decision:**

Reject

**Comment:**

The contributions of this paper are twofold: 1) datasets of tasks are provided, and 2) based on the datasets and hyperparameter lists on the datasets, a transfer learning approach for hyperparameter optimization (HPO) is proposed. Many reviewers positively evaluated the idea and approach discussed in this paper. However, the common concern of multiple reviewers and area chair is that it is not clear whether the provided datasets and their hyperparameter lists are generally applicable to other practical problems. Since there is no discussion on how the datasets are constructed, it is not clear whether they have generally or not. In addition, the comparison with existing HPO approaches is not sufficiently made, and it is not clear whether the performance of the proposed method is advantageous over existing methods. Overall, although the idea is very interesting and potentially useful, I cannot recommend the acceptance in its current form due to the lack of evidence on its generality.